# Increasing large wildfires over the western United States linked to diminishing sea ice in the Arctic

Yufei Zou [1,3✉], Philip J. Rasch [1], Hailong Wang [1✉], Zuowei Xie[2] & Rudong Zhang[1]

The compound nature of large wildfires in combination with complex physical and biophysical processes affecting variations in hydroclimate and fuel conditions makes it difficult to directly connect wildfire changes over fire-prone regions like the western United States (U.S.) with anthropogenic climate change. Here we show that increasing large wildfires during autumn over the western U.S. are fueled by more fire-favorable weather associated with declines in Arctic sea ice during preceding months on both interannual and interdecadal time scales. Our analysis (based on observations, climate model sensitivity experiments, and a multi-model ensemble of climate simulations) demonstrates and explains the Arctic-driven teleconnection through regional circulation changes with the poleward-shifted polar jet stream and enhanced fire-favorable surface weather conditions. The fire weather changes driven by declining Arctic sea ice during the past four decades are of similar magnitude to other leading modes of climate variability such as the El Niño-Southern Oscillation that also influence fire weather in the western U.S.

[1] Atmospheric Sciences and Global Change Division, Pacific Northwest National Laboratory, Richland, WA 99354, USA. [2] International Center for Climate and Environment Sciences, Institute of Atmospheric Physics, Chinese Academy of Sciences, Beijing 100029, China. [3] Present address: Our Kettle, Inc., Kensington, CA 94707, USA. ✉email: yufei.zou@pnnl.gov; hailong.wang@pnnl.gov

Large wildfires are an increasing threat to society and eco-systems over the western U.S., especially across the expanding wildland-urban interface (WUI) regions[1]. Both numbers and total burned areas of large wildfires ($\geq 1000$ acres) in this region have been increasing in the past few decades[2], causing a tremendous socioeconomic burden in terms of devastating casualties and losses, soaring fire prevention and suppression costs, and rising public health risk due to short-term and long-term fire smoke exposure[3,4]. To understand the driving forces of this alarming trend, many studies have investigated its relationship with human activity[5,6] and anthropogenic climate change[7–10] from various perspectives. Unfortunately, the complexity of these interactions among human and natural dimensions of fire activity in the presence of natural variability in the climate system confounds the detection and attribution of changes in large wildfires over this region. Understanding these compound extreme weather events requires a multidisciplinary analysis framework across multiple spatiotemporal scales[11]. Previous studies have suggested that climate impacts on regional fire activity might be masked by human influence on fire ignition and suppression, land use change, and forest management, implying complex human-climate-fire interactions where human impacts often prevail[5,6]. However, for other wildland regions that are less affected by human activity, anthropogenic climate change still exerts predominant impacts on fire by increasing lightning ignitions in boreal forest[12] and permafrost[13] regions, and modulating either fuel availability in resource-limited fire regimes (e.g., xeric shrublands and grasslands/savannas) or fuel aridity in condition-limited fire regimes (e.g., tropical and subtropical forests) where there is ample fuel supply[14]. Although there are multiple fire regimes across the western U.S., several previous studies have found substantial influence of global climate change on increasing forest wildfires over the past four decades through enhanced fuel aridity[8,9] and a reduction of the high-elevation flammability barrier with upslope advance in montane forest fires[15]. Different characteristic scales of anthropogenic and natural processes also shed light on disentangling the fire-centered nexus between human and climate systems for extreme event detection and attribution. For instance, land use change and forest management with excessive fire exclusion and human suppression better explains a forest fire deficit in the western U.S. decoupled from climate and fire weather changes on centennial time scales since the middle 1800s, while the reconciled trends of increasing large fires and worsening fire weather in recent decades suggest an increasingly important role of anthropogenic climate change in modulating regional fire activity on shorter (interannual to interdecadal) time scales[16] as discussed here.

Fire weather variables such as temperature, vapor pressure deficit, and precipitation provide strong explanatory power for seasonal to multidecadal fire activity in empirically based statistical models developed for the western U.S.[8–10] and other fire-prone regions worldwide[14,17–19]. However, a clear causal relationship with mechanistic understanding of complex climate-fire interactions in the Earth system can hardly be drawn from these relatively simple empirical models lack of interacting processes across the Earth's subsystems (e.g., atmosphere, biosphere, and hydrosphere); those causality explanations are better provided by physically-based Earth system models (ESMs) with interactive fire components. In recent years, fire modeling in ESMs has advanced rapidly using various levels of model complexity to represent fire-related climate and vegetation processes[20]. These fire-enabled ESMs provide new tools for investigating the impact of anthropogenic climate change on global and regional fire activity through process-based physical and ecological pathways. For instance, Arctic sea ice has been declining dramatically since the late 1970s particularly in summer and autumn, which is closely related with much stronger warming in the Arctic than the global mean temperature as so-called Arctic amplification (AA)[21]. Several previous studies have suggested significant influence of Arctic sea-ice loss on regional fire weather such as surface air temperature and precipitation based on global and regional climate models[22–25] or even annual wildfire activity in the western U.S. based on statistical analysis[26], but a comprehensive and quantitative evaluation of the Arctic impact on regional burning activity and its role in the observed fire weather change is still lacking. Given increasing but still controversial evidence of emerging connections between high-latitude environmental change and mid-latitude weather extremes in both warm and cold seasons of the past few decades[27–29], further exploration of the potential sea ice-fire teleconnection is worthwhile using latest fire-enabled ESMs.

In this work, we use a series of observation-/reanalysis-/model-based diagnostics and climate model sensitivity experiments to investigate the linkage between declining sea ice in the Arctic and worsening fire hazards in the western U.S. over the past four decades. We first identify an observation-based teleconnection linking regional fire weather and burning activity with Arctic sea-ice changes, and then design and run climate sensitivity experiments using a state-of-the-art fire model with improved fire modeling capability embedded in the Community Earth System model (CESM-RESFire)[30]. This teleconnection is further examined and corroborated across multiple ESMs participated in the latest Coupled Model Intercomparison Project Phase 6 (CMIP6)[31].

## Results

**An observational teleconnection linking regional fire with Arctic sea ice.** Our observation-based statistical analysis suggests a strong negative correlation ($r = -0.68$; $p$-value $< 0.01$) between declining sea-ice concentrations (SIC) over the Pacific sector of the Arctic (120 °E to 135 °W; 70 °N to 80 °N) in preceding summer and autumn (July to October) and worsening fire weather conditions (as described by a Fosberg Fire Weather Index[32], hereafter FFWI, based on the fifth generation of the European Centre for Medium-Range Weather Forecasts (ECMWF) atmospheric reanalysis of the global climate (ERA5)[33]; see Methods section) during the following autumn and early winter (September to December) over the western U.S. (124 °W to 97 °W; 32.5 °N to 48 °N; Fig. 1a, b). Observed changes in composite analyses between years with minimum (hereafter SIC−) and maximum (hereafter SIC+) sea-ice concentrations (marked with triangles in Fig. 1b) in the satellite era further reveal the existence of enhanced fire-favorable regional weather conditions as well as expanded burned area of large wildfires following Arctic sea-ice decline, especially during September to December (Fig. 1c). This correlation is insensitive to the removal of long-term trends in Arctic SIC and regional FFWI ($r = -0.50$ with $p$-value $< 0.01$ after detrending; Supplementary Fig. 1), suggesting a statistically robust linkage between Arctic sea ice and regional fire weather changes across interannual to interdecadal time scales. Results are also statistically significant and robust when other fire weather indices based on different reanalysis data products are used (see Supplementary Discussion and Supplementary Fig. 2). These results raise two questions to be addressed in this work: (1) What is the physical mechanism underlying this robust teleconnection linking worsening regional fire weather and increasing large wildfires with Arctic sea-ice loss across different time scales? (2) How important is this teleconnection effect on regional fire weather changes during the past four decades comparing to the general global warming effect as well as other teleconnections associated with major climate modes such as the El Niño-Southern Oscillation (ENSO)? These two questions are framed under the "Can it?" and "Has it?" analytical framework as

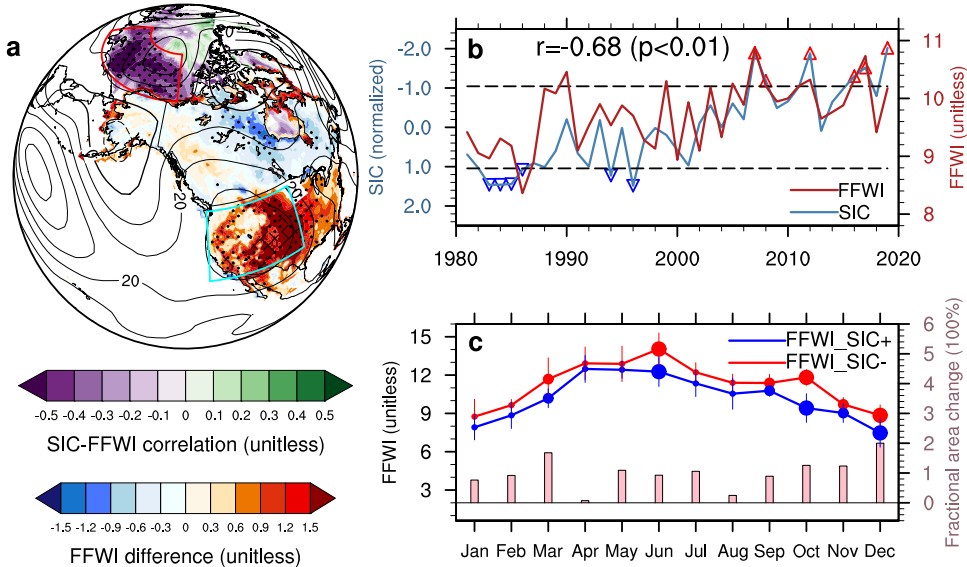

**Fig. 1 Observation- and reanalysis-based teleconnection between Arctic sea ice and regional fire. a** Spatial distributions of the correlation (shading in the Arctic as denoted by the purple-green color bar) between seasonal average Arctic sea-ice concentrations (SIC) in summer and autumn (July to October) and a seasonal and regional average fire weather index (FFWI) over the western U.S. in the following autumn and early winter (September to December), and the difference of seasonal average FFWI (shading in North America as denoted by the blue-red color bar) between the years with minimum (SIC−: red up-pointing triangles in **b**) and maximum (SIC+: blue down-pointing triangles in **b**) Arctic SIC. The difference of seasonal (September to December) average geopotential height at 500 hPa between the SIC− and SIC+ years is also shown in **a** (contours with negative values in dashed lines; unit: m). Stipples in **a** mark regions that are significantly different from 0 at the 0.05 significance level of two-sided $t$-tests, and hatching in **a** denotes statistically significant regions based on the stricter FDR method (see Methods section) with local gridded $p$-value $\leq p_{FDR}^* = 0.0023$ at the threshold of $\alpha_{FDR} = 0.10$. **b** Time series of seasonal and regional average SIC (seasonal mean from July to October; normalized by its 1981–2010 climatological mean and standard deviation; note its scale on the left $Y$-axis is inverted to directly compare temporal variations of both time series), FFWI (seasonal mean from September to December), and their correlation. The region definitions for the Pacific sector of the Arctic and the western U.S. are outlined by the red and cyan boxes in **a**, respectively. The horizontal dashed lines denote the ±1 standard deviations of normalized SIC as thresholds for selecting the SIC± years. **c** The composite of monthly FFWI (solid lines with dots and error bars) and fractional burned area change of large wildfires (vertical bars) over the western U.S. Error bars in **c** denote ±1 standard deviations of monthly FFWI in each group. Dot sizes for monthly FFWI in **c** Denote the 0.05 (large),0.1 (medium), and non-significant (small) significance levels of two-sided $t$-tests for monthly FFWI differences between the years with minimum (FFWI_SIC−) and maximum (FFWI_SIC+) Arctic SIC, respectively.

suggested by a previous study[34] discussing how to avoid ambiguities about the climate impacts of Arctic warming.

**Increased regional fire in response to Arctic sea-ice loss in CESM-RESFire**. To answer the first question, we have designed and conducted two CESM-RESFire sensitivity experiments by replacing the climatological Arctic sea-ice concentrations and associated sea surface temperature (SST) from July to October in the 40-year control run with the multi-year average Arctic SIC/ SST conditions corresponding to the observed minimum (hereafter SICexp−) and maximum (hereafter SICexp+) SIC years to isolate the impact of preceding Arctic sea-ice loss on regional fire weather and burning activity in the following autumn and early winter (September to December; see Methods section). The modeling results show an anomalous dipole pattern in the 500 hPa geopotential height field averaged over September-December (hereafter Z500) with cyclonic (negative) anomalies centered over Alaska and anticyclonic (positive) anomalies centered over the western U.S. in response to prescribed sea-ice reduction in preceding July to October (Fig. 2a). This anomalous circulation pattern is similar to that revealed by differencing ERA5 reanalysis-based composites between the SIC− and SIC+ years (Fig. 1a). The similarity is even more striking over the downstream North America continental regions when the SIC, FFWI, and burned area time series as well as gridded reanalysis data are all detrended before differencing the new groups of the years with minimum (hereafter SICnotrd−) and maximum

(hereafter SICnotrd+) sea-ice concentrations in the detrended SIC time series (Supplementary Fig. 1b; see Methods section). Considering that the long-term trends in these variables are closely related with global warming effects, the improved similarity between the modeling results and the detrended reanalysis data is understandable because of the absence of global warming effects in the SICexp− and SICexp+ simulations other than those deliberately exposed to the Arctic sea ice-driven local warming effect. Since the sea-ice forcing used in the climate model sensitivity experiments results from both interannual and interdecadal changes, we use the original sea ice and reanalysis data affected by both short-term and long-term variability and the detrended data mainly affected by interannual variability for the following composite analyses to test the robustness of the composite results on different time scales. Note that the SIC± (Fig. 1b) and SICnotrd−/notrd+ (Supplementary Fig. 1b) years are different for the two observation-/reanalysis-based composite analyses because of different SIC time series used for the composite member selection.

We project the simulated Z500 anomalies onto to a fire-favorable regional circulation pattern to obtain its corresponding time series (hereafter Z500i) in each experiment (see Methods). The statistical distributions of Z500i suggest a positive shift in SICexp− (Fig. 2b; $p$-value = 0.01 based on a two-sided $t$-test for the 40 paired Z500i samples from the SICexp± experiments; the same test is also used for the following FFWI, burned area, fire occurrence, and fire size comparisons between the SICexp± experiments) concurrent with other regional weather changes including suppressed clouds and precipitation and increased incoming solar radiation over the

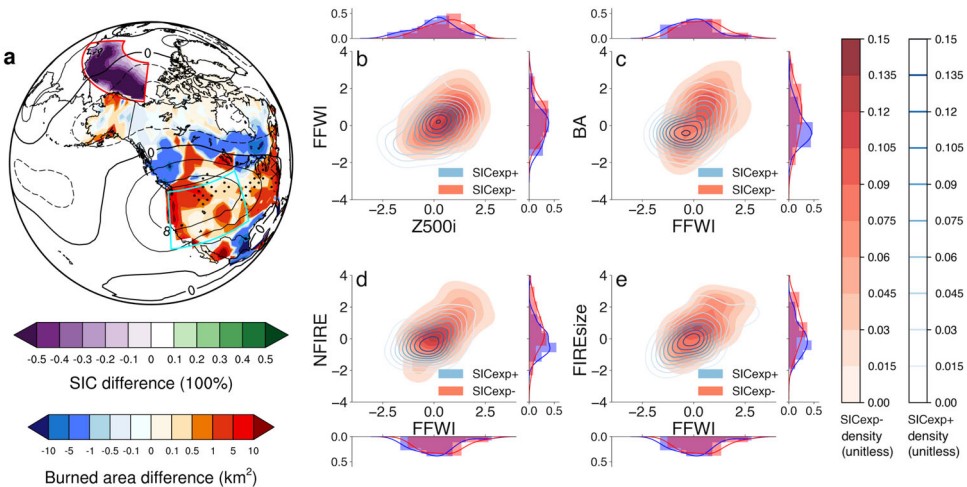

**Fig. 2 CESM-RESFire simulated Arctic sea ice and regional fire teleconnection. a** Spatial distributions of the seasonal average (July to October) sea-ice concentration (SIC; unit: 100%) difference (color shading in the Arctic Ocean as denoted by the purple-green color bar) between the SICexp− and SICexp+ experiments, and the seasonal average (September to December) burned area difference (color shading in North America as denoted by the blue-red color bar) in response to the sea-ice perturbation. The difference of geopotential height at 500 hPa (Z500; unit: m) between SICexp− and SICexp+ is also shown (contours with negative values in dashed lines). The region definitions for the Pacific sector of the Arctic and the western U.S. are outlined by the red and cyan boxes, respectively. Stipples in **a** show regions that are significantly different from 0 at the 0.1 significance level of two-sided $t$-tests, and hatching in **a** denotes statistically significant regions based on the stricter FDR method (see Methods section) with local gridded $p$-value $\leq p^*_{FDR} = 0.017$ at the threshold of $\alpha_{FDR} = 0.20$. **b** Two-dimensional joint distributions of the seasonal mean fire-favorable circulation index (Z500i; standardized by first removing the 40-year mean and then normalizing by the standard deviation of Z500i from the SICexp+ experiment; unitless) and fire weather index (FFWI; also standardized by the 40-year mean and standard deviation of FFWI from the SICexp+ experiment; unitless) based on the kernel density estimation (KDE) for SICexp− (red shading) and SICexp+ (blue contours). The legends for color shading and contours are attached aside, and corresponding 1-d KDE distributions for each index in SICexp− (red) and SICexp+ (blue) are also shown along the x- and y-axis. **c** As in **b**, but for the comparison of FFWI and regional total burned area (BA; unitless) that are both standardized. **d** As in **b**, but for the comparison of FFWI and regional mean fire occurrence (NFIRE; unitless) that are both standardized. **e** As in **b**, but for the comparison of FFWI and regional mean fire size (FIREsize; unitless) that are both standardized. Note that the model-based fire variables are averaged over the coarse model grid cells that describe statistical properties of fire ensembles at each grid cell rather than individual properties of each single fire.

western U.S. (Supplementary Fig. 3), contributing to a positive shift of regional FFWI (Fig. 2b; $p$-value = 0.06) with more frequent and intensified hot and dry surface weather conditions during autumn and early winter in SICexp−. Accordingly, such fire-favorable weather in SICexp− is conducive to more extensive burning activity over the western U.S. as suggested by expanded regional burned area (Fig. 2a, c; $p$-value = 0.04) due to both increased fire occurrence (Fig. 2d; $p$-value = 0.04) and enlarged fire size (Fig. 2e; $p$-value < 0.01). A month-by-month comparison between SICexp− and SICexp+ also shows consistent changes in regional fire weather, fire occurrence, fire size, and total burned area in consecutive months after Arctic sea-ice declining, with the largest increase of regional total burned area by ~12.5% in November (Supplementary Fig. 4). Besides the ensemble mean responses, we also examine the probability and intensity changes of extreme burning years (defined by modeling years in each experiment with regional and seasonal total burned area above the 95% percentile of the SICexp+ results) using a bootstrap resampling method (see Methods section). The results show dramatic increases with nearly four times higher occurrence probability and 14–15% higher burning intensity of extreme burning years under the SICexp− condition than that under the SICexp+ condition (Supplementary Fig. 5). This significant increase in the occurrence probability of extreme burning years is robust for both bootstrapping estimates without and with sample replacement once the modeling ensemble size exceeds 30 years (Supplementary Fig. 6).

**Consistent fire weather changes in the reanalysis and modeling results.** Additional diagnostics of atmospheric dynamics and thermodynamics help to better understand the physical processes contributing to the above sea ice-driven fire expansion. The reanalysis-based difference in zonally averaged temperature (from the ERA5 reanalysis product) between SIC− and SIC+ years shows strong but heterogeneous warming both near the surface and in the free troposphere over mid- and high-latitude regions, manifested by an enhanced (reduced) meridional temperature gradient ~60 °N (80 °N) in the lower and middle troposphere (Fig. 3a). This feature of an increased baroclinity around 60 °N is also evident in the composite difference between SICnotrd− and SICnotrd+ years based on the detrended ERA5 reanalysis data (Fig. 3b), which is well captured by the model sensitivity experiments (Fig. 3c) through atmospheric dynamics-driven processes rather than from physical processes such as diabatic heating or vertical diffusion (Supplementary Fig. 7). The warming magnitudes in the detrended ERA5 data and model simulations are weaker than that in the original ERA5 data because the long-term global warming trend is absent in the detrended data and model simulations; moreover, the SST distributions outside the perturbed Arctic region in the model simulations are identical and the climate forcing agents such as greenhouse gases (GHGs) and aerosols repeat the same climatological cycle each year in both SICexp− and SICexp+ experiments so as to isolate the modeled response to regional Arctic sea-ice reduction and local SST warming. Nevertheless, the meridional temperature structures showing similar temperature gradient patterns correspond to similar circulation changes in zonally averaged zonal wind through the thermal wind relation, manifesting a poleward shift of the polar jet stream and storm tracks in all three reanalysis- and model-based datasets (Fig. 3d–f). These poleward shifts are also evident in horizontal wind and precipitation fields, resulting in a wetter Pacific Northwest coast and most inland regions in Alaska and Canada but a drier western and midwestern U.S. (Fig. 3g–i). These changed hydroclimate conditions between SIC− and SIC + years are consistent across

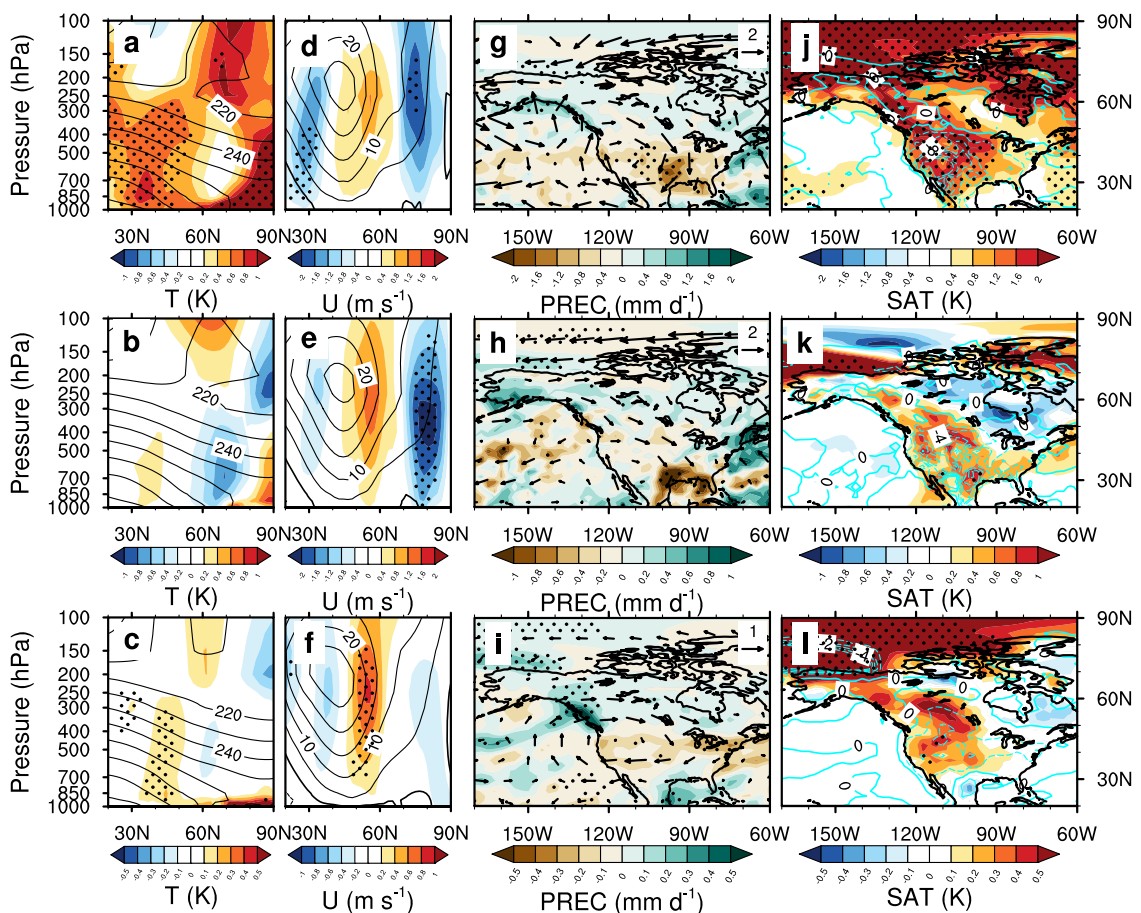

**Fig. 3 Physical processes underlying the Arctic sea ice and regional fire teleconnection. a** Zonally averaged (170 °W to 60 °W; as shown in **g**) temperature (T; color shading; unit: K) difference in autumn and early winter (September to December) between the years with minimum (SIC−) and maximum sea-ice concentration (SIC+) based on the original ERA5 reanalysis data. The time average of zonally averaged temperature in the SIC+ years is also shown (contours; unit: K). **b** As in **a**, but for the temperature difference between the years with minimum (SICnotrd−) and maximum sea-ice concentration (SICnotrd+) based on the detrended ERA5 reanalysis data. **c** As in **a**, but for the temperature difference between the experiments with minimum (SICexp−) and maximum sea-ice concentration (SICexp+) based on the CESM-RESFire simulations. **d–f** As in **a–c**, but for zonally averaged zonal wind (U; color shading; unit: m s⁻¹) difference based on the original ERA5 reanalysis data, the detrended ERA5 reanalysis data, and the CESM-RESFire simulations, respectively. **g–i** As in **a–c**, but for wind circulation at 500 hPa (arrows; unit: m s⁻¹) and total precipitation rate (PREC; color shading; unit: mm d⁻¹) differences based on the original ERA5 reanalysis data, the detrended ERA5 reanalysis data, and the CESM-RESFire simulations, respectively. **j–l** As in **a–c**, but for surface relative humidity (cyan contours with negative values in dashed lines; unit: %) and surface air temperature (SAT; color shading; unit: K) differences based on the original ERA5 reanalysis data, the detrended ERA5 reanalysis data, and the CESM-RESFire simulations, respectively. Stipples in **a–l** show regions that are significantly different from 0 at the 0.1 significance level of a two-sided *t*-test.

different observational and reanalysis precipitation datasets other than the one used here (Supplementary Fig. 8). The associated anticyclonic circulation anomaly over the western U.S. also suppresses cloud formation with enhanced downward motion and incoming solar radiation, resulting in hotter and drier surface weather that is synergistic for enhanced fuel aridity across the western U.S. (Fig. 3j–l). The consistency between the original and detrended reanalysis data suggests a robust dynamical linkage between the Arctic sea ice and regional fire weather on both long-term and short-term time scales, while the resemblance between the model and the reanalysis-based results confirms the significant impact of declining Arctic sea ice on exacerbated burning activity through modulating regional circulation and surface fire weather conditions.

**Substantial Arctic teleconnection effects across multiple CMIP6 models.** The synergistic effects of these multivariate fire weather changes following pre-conditioned peak fire season during climatologically hot and dry summer prolongs regional

fire season and exacerbates burning severity in the subsequent autumn and early winter following diminished Arctic sea ice, making the prediction of fire hazards well-suited to an analysis of compound climate extreme events[35]. However, the complexity of multiple interactive pathways and processes in climate systems and relatively short observational records also increase the difficulty of detection and attribution of climate extreme events in the observed historical data. As mentioned above, the model-simulated warming and associated hydroclimate anomalies are generally weaker than the reanalysis-based composite difference because of the fixed climate forcing agents such as GHGs and extra-polar SST changes other than the local warming effect induced by Arctic sea-ice loss in the modeling sensitivity experiments. The missing global warming effect in the atmosphere and extra-polar oceans in the CESM modeling settings limits its capability to answer the second question about the importance of the Arctic-driven teleconnection effect in the observed fire weather changes associated with all climate variability and interactive processes.

To improve the robustness of our analysis and preclude the potential influence of other processes and pathways driven by variable climate forcing agents (e.g., increasing GHGs in the past four decades) on historical fire weather variability, we also examine the difference of climate responses between the Atmospheric Model Intercomparison Project (amip)[31] and its counterfactual counterpart with pre-industrial forcing (amip-piForcing or AMIP SSTs with control forcing)[36] experiments in CMIP6 (see Methods). These two experiments share the same realistic and observationally-based SST and sea-ice surface conditions from 1979 to near present-day but expose model sensitivity to different anthropogenic forcing levels of GHGs, aerosols, and land use change by simulating the model response to time varying realistic forcing levels (as in the amip scenario) and to forcing levels held constant at pre-industrial levels (as in the amip-piForcing scenario). The differences between the SIC− and SIC+ years in both amip (Supplementary Fig. 9a–d) and amip-piForcing (Supplementary Fig. 9e–h) experiments agree with each other with minor changes in fire-related climate responses, suggesting a dominant role of ocean/sea-ice surface conditions in driving the observed fire weather changes (i.e., fire weather changes over the U.S. resemble each other in the amip and amip-piForcing experiments despite different levels of atmospheric and terrestrial forcing agents). The difference between these two experiments further reveals mixed climate effects of anthropogenic and natural forcing through atmospheric and land processes on regional fire weather changes. Specifically, only the response in warmer surface air temperature (Supplementary Fig. 9l) over the continental U.S. is consistent with the reanalysis-based composite result (Fig. 3j), while the climate responses to the airborne and terrestrial anthropogenic forcing (i.e., GHGs, aerosols, and land use change) reflected in zonally averaged temperature (Supplementary Fig. 9i) and wind (Supplementary Fig. 9j) fields as well as horizontal wind circulation and precipitation (Supplementary Fig. 9k) show distinct signatures that differ from the reanalysis-based results (Fig. 3a, d, g). These results indicate that the observed regional fire weather changes between the SIC− and SIC+ years are strongly controlled by oceanic surface conditions including both sea-ice and SST changes in a warming climate rather than by a direct response through atmospheric or terrestrial processes to the climate forcing agents.

The SST and sea-ice changes appear to be the critical ingredients in eliciting the regional fire weather conditions conducive to more large wildfires–but which one is more important? We further conduct a pattern recognition method known as the "signal-to-noise-maximizing pattern (S/NP) filtering method"[37] based on the ERA5 reanalysis and the amip model ensemble to separate forced responses in regional fire weather due to other climate variability such as tropical ocean variations associated with ENSO (see Methods). The multi-field pattern filtering results show ENSO- and Arctic-driven hemispherical teleconnection patterns emerging in the first (S/NP1; Supplementary Fig. 10) and third (S/NP3; Supplementary Fig. 11) groups of S/NPs, respectively. These two groups of S/NPs show consistent responses in regional fire weather with predominant contributions to a warmer and drier western U.S. during the SIC− years by similar magnitudes. Repeating this analysis using the detrended ERA5 reanalysis and amip-piForcing data shows similar results (e.g., a hotter and drier western U.S. during the SICnotrd− years) that are also constructively contributed by interannual variations of both ENSO (Supplementary Fig. 12) and Arctic sea ice (Supplementary Fig. 13) even when the long-term global warming and AA effects have been removed in the detrended data. Therefore, these climate diagnostic results along with our Arctic sea ice sensitivity experiments support the hypothesis that Arctic surface conditions play an important and synergistic role in

determining regional fire weather and burning activity changes over the western U.S. across interannual to interdecadal scales. Given the continuously increasing trend in the S/NP3 time series and decreasing trend in Arctic sea ice (Supplementary Fig. 11), Arctic-driven teleconnection effects are expected to play an increasingly prominent role in modulating regional fire weather in the future.

The hypothesized Arctic-driven teleconnection has been further examined using the amip simulations to corroborate the robustness of these relationships across different amip participating models. Since the amip experiment uses the observational SST and sea ice surface boundary conditions, the time series of SIC in the amip model ensemble is almost identical to the observational data (Fig. 4a). The ensemble simulated atmospheric and land surface fire weather variations in terms of Z500i ($r_{amip-ERA5}^{Z500i} = 0.74$; $p$-value < 0.01) and FFWI ($r_{amip-ERA5}^{FFWI} = 0.54$; $p$-value < 0.01) also reproduce the reanalysis-based interannual and interdecadal variations in general, with comparable correlation coefficients among SIC, Z500i, and FFWI between the amip model ensemble and the reanalysis data (Supplementary Fig. 14). All 15 models capture the correct signs of correlations between SIC and Z500i as well as Z500i and FFWI, respectively, and a majority of participating models (12 out of 15) successfully reproduce the negative correlation between SIC and FFWI with 5 of them showing statistically significant negative correlations like that found in the reanalysis data (Supplementary Fig. 14). We then analyze the spatial distributions in the 12 amip models showing correct SIC-FFWI correlations by comparing the differences of non-filtered (Fig. 4b, c) and S/NP3 filtered patterns (Fig. 4d, e) in precipitation and FFWI between the SIC− and SIC+ years to tease out the role of the Arctic-driven fire weather changes in all forcing-driven changes. The changes in S/NP3 filtered fields appear to be mostly driven by the Arctic change, with little forced response evident from other climate drivers. Though the magnitudes of the Arctic-driven fire weather changes in precipitation (Fig. 4d) and FFWI (Fig. 4e) are about half of the corresponding total changes (Fig. 4b, c), the north-south contrast spatial patterns are well preserved in the S/NP3 filtered fields with severely deteriorated fire weather occurring over the western U.S. during SIC− years. Moreover, the changes in regional fire weather (i.e., precipitation; FFWI) associated with the Arctic and ENSO variability are comparable and constructive with each other, contributing to synergistical enhancement in their net effects by adding them together that resemble the total changes driven by all forcing and variability (Supplementary Fig. 15). The lower ranking and larger ensemble spread of S/NP3 (Supplementary Fig. 11) than S/NP1 (Supplementary Fig. 10) suggest a lower signal-to-noise ratio of Arctic-driven teleconnection effects than ENSO-driven effects in the amip models, which is partly responsible for the controversial role of high-latitude changes in middle-latitude climate and weather extremes owing to diversified climate modeling responses to Arctic sea-ice loss[29,38]. Improvement on the representation of Arctic-mid-latitude teleconnection in ESMs might pose another grand challenge to climate model development because of all the complex dynamic and physical processes involved in the long teleconnection pathway across multiple components of the Earth system.

## Discussion

Recently, significant progress has been made to understand the linkage between high-latitude climate change and mid-latitude weather extremes, but there is a lack of consensus among the community about the potential mechanisms due to the relatively short length of observational datasets and low signal-to-noise ratios in climate modeling results[29]. Most previous studies have focused on the Arctic influence on climate and weather extremes in winter[27] or summer[28], with compound extreme events during

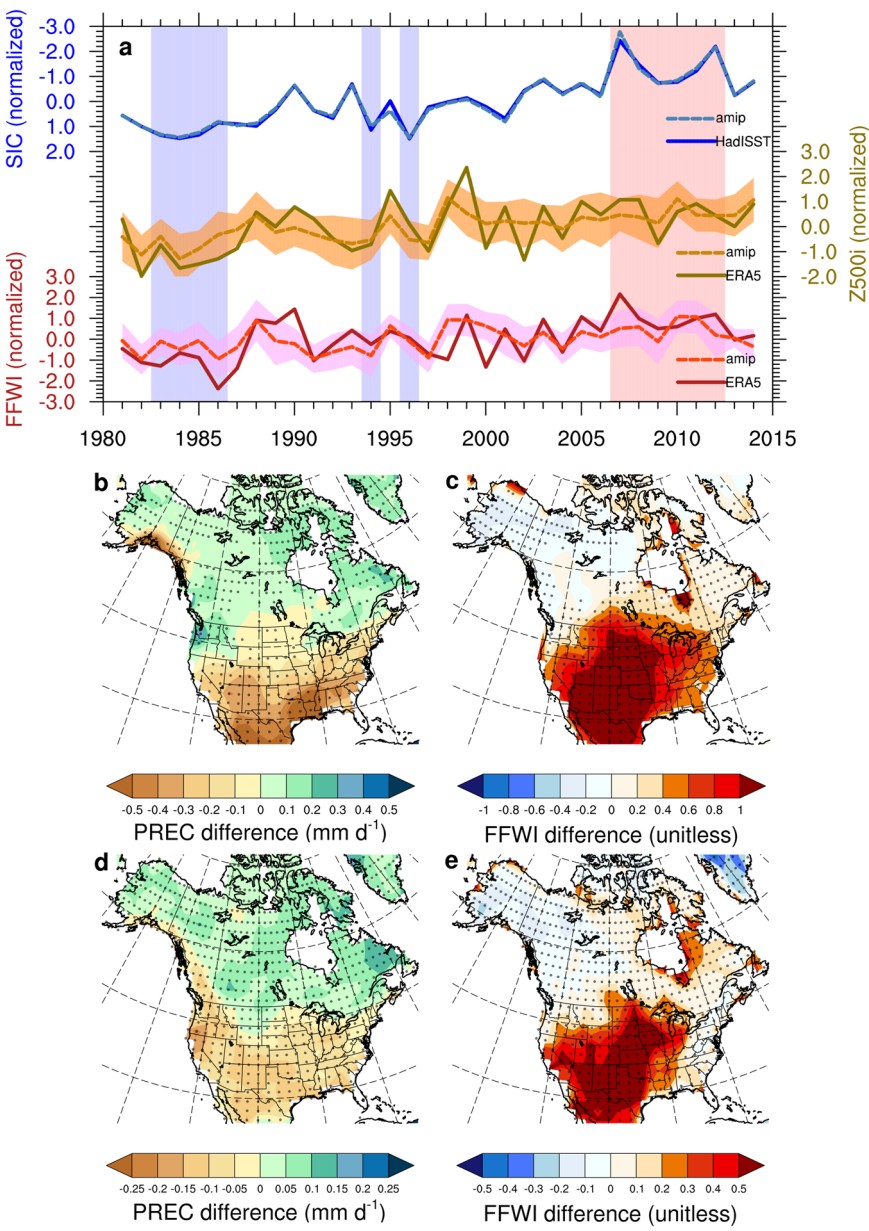

**Fig. 4 Arctic sea ice and regional fire weather teleconnection in the CMIP6 amip experiment. a** Time series of normalized seasonal and regional average sea-ice concentration (SIC; unitless), a fire-favorable circulation index (Z500i; unitless), and a fire weather index (FFWI; unitless) based on the observational/reanalysis data and 15 amip model ensemble. The shading along lines of the time series denotes ±1 standard deviations of the model ensemble, and the vertical bar shading denotes years with minimum (pink; SIC−) and maximum (blue; SIC+) SIC for the composite differences in **b** and **c**. Note the SIC scale on the top left Y-axis is inverted for direct comparison with the other two variables. **b** The total difference in precipitation rates (PREC; color shading; unit: mm d⁻¹) in autumn and early winter (September to December) between the SIC− and SIC+ years based on the 12 amip model ensemble. **c** As in **b**, but for the total difference in the fire weather index (FFWI; color shading; unitless). **d** The Arctic-driven (S/NP3) changes in precipitation rates (PREC; color shading; unit: mm d⁻¹) in autumn and early winter (September to December) between the SIC− and SIC+ years based on the 12 amip model ensemble. **e** As in **d**, but for the Arctic-driven (S/NP3) changes in the fire weather index (FFWI; color shading; unitless) based on the 12 amip model ensemble. Stipples in **b**–**e** show regions that 2/3 amip models agree on the signs.

transitional seasons such as large wildfires in autumn receiving less attention. The competition of different dynamic pathways and processes[39] in those transitional seasons increases the difficulty of obtaining a clear view (with consensus) about the climate influence on compound extreme events such as the fire hazard discussed here. Our combined analyses based on both single forcing-oriented climate sensitivity experiments and all forcing-included ensemble historical modeling experiments enable us to understand the physical mechanism underlying the statistical relationship as well as its relative contributions to the observed

fire weather changes. Massive Arctic sea-ice loss in summer and autumn strongly increases surface absorption of incoming solar radiation through the positive surface albedo feedback, resulting in a strong surface warming and strengthened upward turbulent heat fluxes into the lower troposphere over the sea-ice melted Arctic regions. This surface warming anomaly induces a cyclonic potential vorticity anomaly[40] over the heating source and vicinity land regions like Alaska, which further enhances warm air advection from the ocean into downstream land regions like the western U.S. This dynamic-driven warming anomaly over the

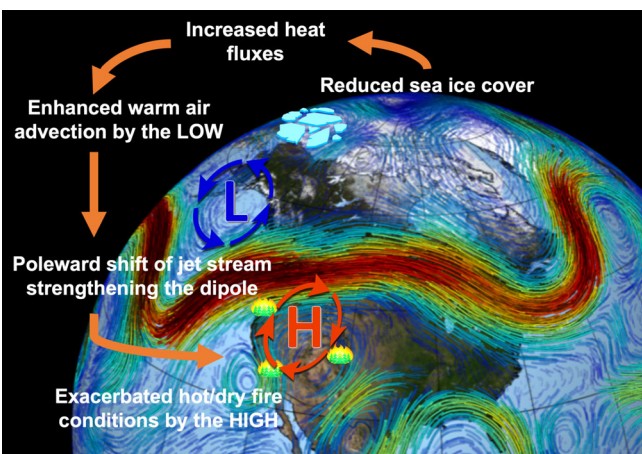

**Fig. 5 A schematic diagram for the teleconnection between Arctic sea-ice loss and increasing fire hazards over the western U.S.** The L and H denote the cyclonic low pressure and anticyclonic high pressure circulation anomalies, respectively, induced by preceding Arctic sea-ice loss as suggested by the CESM-RESFire model sensitivity results shown in Fig. 2a. (Background image by NASA/Goddard Space Flight Center Scientific Visualization Studio).

western U.S. helps to develop and strengthen an anticyclonic anomaly in the lower to middle troposphere of this region that manifests as a dipole pattern in conjunction with the cyclonic anomaly over the upstream regions like Alaska. This dipole pattern corresponds to a poleward-shifted polar jet stream with suppressed precipitation and elevated surface air temperature and vapor pressure deficit over the western U.S., all conducive to increased fuel aridity and more extensive burning during the following autumn and early winter in this region (Fig. 5).

Although some interactive processes such as ocean-atmosphere coupling and fire-climate feedbacks are missing in the modeling experiments used in this study, previous studies have investigated the possible influence of those interactive processes and suggested consistent but amplified climate responses through ocean-atmosphere coupling[41] and nonnegligible but secondary fire-climate feedbacks through land-atmosphere coupling[42]. Moreover, more coordinated climate modeling intercomparison projects such as the Polar Amplification Model Intercomparison Project (PAMIP)[43] and the Fire Model Intercomparison Project (FireMIP)[44] have been designed and conducted, which help narrowing the knowledge gaps regarding the teleconnection linking high-latitude and mid-latitude regions and its role in increasing fire hazards.

More extreme fire weather with increasing likelihood of large wildfires in autumn has become the new normal for western regions like California, a region projected to suffer more by the end of this century as has the clear decline in Arctic sea-ice coverage[45]. Previous studies have identified strategies for coexistence with wildfires in a changing climate with escalating fire danger[46,47]. But Arctic sea ice has been projected to continuously decline and eventually diminish to a sea ice-free Arctic in September before the 2050s[48], so more drastic changes might be anticipated. This study describes a mechanism indicating how the teleconnection between decreasing Arctic sea ice and worsening regional fire weather may be sustained and even strengthen over the next few decades, favoring more and larger wildfires across the western U.S. and making this region, especially the growing WUI areas, even more susceptible to destructive fire hazards. These implications may serve as motivation for more attention to adaptive resilience approaches including public awareness of fire risk and hazard mitigation, scientific fire risk and forest

management, and sustainable residential and infrastructure development planning on fire-prone landscapes[46,47].

## Methods

**Observation and reanalysis data.** The $1° × 1°$ gridded sea-ice concentrations at monthly frequency for 1981–2019 are provided by the Hadley Centre Sea Ice and Sea Surface Temperature dataset (HadISST)[49]. Daily and monthly meteorological variables including air temperature, 2-m relative humidity, wind speed and vectors, total precipitation rates, and geopotential heights used for the FFWI calculation and fire weather composite analysis are collected and processed based on the ERA5 reanalysis dataset[33]. We also analyzed the Canadian Forest Fire Weather Index (FWI)[50] datasets that are calculated based on the ERA5 reanalysis data[33] from the Global ECMWF Fire Forecast model (GEFF-ERA5)[51] and the NASA's Modern-Era Retrospective Analysis for Research and Applications (MERRA-2)[52] reanalysis data from the Global Fire Weather Database (GFWED)[53], respectively, as well as multiple observation- and reanalysis-based precipitation datasets from the Global Precipitation Climatology Project (GPCP v2.3)[54], MERRA-2[52], Climate Forecast System Reanalysis (CFSR)[55], ERA-Interim[56], and the Japanese 55-year Reanalysis (JRA-55)[57] for testing the robustness of the Arctic sea ice-fire teleconnection in Supplementary Table 1, Supplementary Fig. 2, and Supplementary Fig. 8.

The observational dataset of large wildfires ($≥1000$ acres) across the western U.S. from 1984 to present are produced by the U.S. Geological Survey Center and the USDA Forest Service through the Monitoring Trends in Burned Severity (MTBS) program[58]. We update the relative old MTBS fire data used in Dennison et al.[2] and show the continuously increasing trends in both numbers and burned areas of large wildfires over the western U.S. in Supplementary Fig. 16. The fractional changes of regional total burned area (BAC) in Fig. 1c, Supplementary Fig. 1c, and Supplementary Fig. 2c are given by:

$$ \text{BAC} = \frac{\overline{\text{BA}}_{\text{SIC}-} - \overline{\text{BA}}_{\text{SIC}+}}{\frac{1}{2} \left( \left| \overline{\text{BA}}_{\text{SIC}-} \right| + \left| \overline{\text{BA}}_{\text{SIC}+} \right| \right)} \tag{1} $$

where $\overline{\text{BA}}_{\text{SIC}-}$ and $\overline{\text{BA}}_{\text{SIC}+}$ are the monthly total burned area of large wildfires averaged over the minimum and maximum SIC years in the original (SIC−/SIC+; as shown in Fig. 1b) or detrended (SICnotrd−/SICnotrd+; as shown in Supplementary Fig. 1b) SIC time series, respectively. Note that the MTBS dataset starts from 1984 with a shorter time period than the SIC time series, so the group members for $\overline{\text{BA}}_{\text{SIC}+}$ are slightly different (with missing years before 1984) from the composite analyses for other variables like FFWI. We choose the average of $\left| \overline{\text{BA}}_{\text{SIC}-} \right|$ and $\left| \overline{\text{BA}}_{\text{SIC}+} \right|$ rather than $\left| \overline{\text{BA}}_{\text{SIC}+} \right|$ alone as the denominator in case of zero $\left| \overline{\text{BA}}_{\text{SIC}+} \right|$ in some months such as December or negative burned area anomalous values after detrending in SICnotrd− and SICnotrd+ years. In these cases, BAC becomes 200% or −200% according to Eq. (1) (e.g., Dec in Fig. 1c; May–Jun and Sep–Nov in Supplementary Fig. 1c).

**Models and experiments.** We use a process-based CESM-RESFire model[30] for the Arctic sea-ice climate sensitivity experiments. The RESFire model was developed with improved region-specific fire weather and socioeconomic constraints and fire feedbacks to the climate and vegetation, and coupled with both the land and the atmosphere components of the CESM version 1.2 modeling system[59]. The major new features of this fire model include online coupled fire emissions in forms of mass and energy fluxes with fire plume rise and associated radiative effects as well as fire-induced land cover change and disturbances to terrestrial biogeochemical cycle. The land component we use for the sensitivity experiments is the Community Land Model version 4.5 (CLM4.5)[60], while the atmosphere component we use is the high-top Whole Atmosphere Community Climate Model (WACCM)[61], which is a comprehensive atmospheric model with a well-resolved stratosphere by 70 vertical levels up to 140 km at a horizontal resolution of $1.9°$ (latitude) $× 2.5°$ (longitude).

We first conduct a 40-year simulation as a control (CTRL) run with annually repeating prescribed climatological (1981–2010 average) SIC and SST from the HadISST dataset[49] to generate initial conditions for two sensitivity runs. We then branch two climate sensitivity experiments (i.e., SICexp+ and SICexp−) from July to December of each modeling year by perturbing SIC and SST in the Pacific sector of the Arctic (120 °E to 135 °W; 70 °N to 80 °N; as shown in Fig. 2a) from July to October to investigate the climate response to regional Arctic SIC and associated local SST changes. The selection of the perturbation months is based on the correlation coefficients of seasonal FFWI and monthly Arctic SIC of the same years that are statistically significant with both original and detrended monthly SIC time series. In the SICexp+ experiment, we replace the climatological SIC and SST in the selected Arctic region with the averaged SIC and SST over the six maximum SIC years (1983, 1984, 1985, 1986, 1994, and 1996; as shown in Fig. 1b) above positive one standard deviation ($> +1σ$) of the normalized regional SIC during 1981–2019. In the SICexp− experiment, we replace the climatological SIC and SST in the selected Arctic region with the averaged SIC and SST over the six minimum SIC years (2007, 2008, 2012, 2016, 2017, 2019; as shown in Fig. 1b) below negative one standard deviation ($< −1σ$) of the normalized regional SIC during 1981–2019. All other conditions such as the initial conditions, extra-polar SST boundary conditions, and lightning and population density for natural and anthropogenic fire ignition are kept the same in both experiments (Supplementary Table 2).

Finally, we examine the simulated responses in fire weather (e.g., T, Q, and U10 for FFWI calculation; the sum of PRECC and PRECL for precipitation) and burning variables (e.g., FAREA_BURNED for fractional burned area; NFIRE for fire counts; BA_AVG for mean fire size at each model grid cell) in terms of changes in spatial/temporal patterns and statistical distributions between the SICexp− and SICexp+ experiments. This method has been applied in similar model sensitivity studies on climate impacts of sea-ice loss[62,63].

To focus on the sea-ice influence on fire, we turn off the fire feedback processes including online coupled fire emissions and fire-induced land cover change in the RESFire model. These feedback processes would exert detectable but secondary effects on global burning activity by either amplifying or damping regional burned area simulations as suggested by the comprehensive evaluation in our previous study[42]. For the western U.S. region of interest in this study, the net fire feedback effect is almost negligible as shown in the Supplementary Fig. 17, which implies that the simplified modeling setting without consideration of fire feedbacks in this study is appropriate to meet its major research objective.

Besides the CESM-RESFire climate sensitivity experiments, we also examine the Arctic sea ice-fire teleconnection in multiple CMIP6 climate modeling systems participating in the amip[31] and amip-piForcing[36] experiments. There are 15 CMIP6 models (Supplementary Table 3; hereafter amip models) in total that provide the daily and monthly model outputs of the amip experiment for the FFWI calculation and fire weather analysis in Fig. 4, and 4 CMIP6 models (Supplementary Table 4; hereafter amip-piForcing models) provide the monthly model outputs of the amip-piForcing experiment for the fire weather composite analyses in Supplementary Figs. 9, 12, and 13. Most model outputs in both amip and amip-piForcing experiments end in 2014, so the composite years for the SIC− group (2007–2012; as shown in Fig. 4a and Supplementary Figs. 10c, 11c) in these models are slightly different from those in the ERA5-based composite analysis (the composite years for the SIC+ group are the same as shown in Fig. 4a). For the S/NP filtering analyses based on detrended amip-piForcing data, all the composite years for both SICnotrd− (1981, 1990, 1993, 2007, 2008, 2012; as shown in Supplementary Figs. 12c 13c) and SICnotrd+ (1994, 1996, 2000, 2001, 2013; as shown in Supplementary Figs. 12c and 13c) groups are also slightly different from those in the ERA5-based SICnotrd− (with one more year of 2019) and SICnotrd+ (with one more year of 2018) groups in Supplementary Fig. 1b.

**Climate and weather indices.** FFWI[32] is an empirical fire weather index provided as a complement of the National Fire Danger Rating System (NFDRS)[64] for measuring the weather influence on fire danger in the U.S. It is calculated based on surface air temperature, 2-m relative humidity, and surface wind speed with larger values implying higher fire risk or potential. FFWI is given by:

$$\text{FFWI} = \frac{\eta\sqrt{1+U^2}}{0.3002} \tag{2}$$

where $U$ is surface wind speed in miles per hour and $\eta$ is the moisture damping coefficient.

$$\eta = 1 - 2\times\left(\frac{\text{EMC}}{30}\right) + 1.5\times\left(\frac{\text{EMC}}{30}\right)^2 - 0.5\times\left(\frac{\text{EMC}}{30}\right)^3 \tag{3}$$

where EMC is the equilibrium moisture content as a function of surface air temperature $T$ in degrees Fahrenheit and relative humidity RH in percentage:

$$\text{EMC} = \begin{cases} 0.03229 + 0.281073\times\text{RH} - 0.000578\times T\times\text{RH}, \text{ for RH} < 10\% \\ 2.22749 + 0.160107\times\text{RH} - 0.01478\times T, \text{ for } 10\% < \text{RH} \leq 50\% \\ 21.0606 + 0.005565\times\text{RH}^2 - 0.00035\times T\times\text{RH} - 0.483199\times\text{RH}, \text{ for RH} > 50\% \end{cases} \tag{4}$$

This EMC variable is the same with that used in NFDRS as the fundamental basis to all fuel moisture computations in that more complicated system. All the calculations of FFWI only require meteorological values at observational time or corresponding modeling outputs. It has been previously used for evaluation of climate change impacts on wildfire in California[10] and weekly to seasonal fire danger forecasts in the U.S.[65]. In comparison, FWI[50] is a more complicated system based on multiple fuel moisture codes and fire behavior indices, which are derived from meteorological variables including surface air temperature, relative humidity, wind, and precipitation at local noon time. We choose FFWI in combination with precipitation rather than FWI for fire weather analysis in the main text because of its less stringent requirement for meteorological data inputs given limited weather/fuel data availability from daily-based climate model outputs. Using different fire weather indices or reanalysis datasets for analysis does not change the major finding of the identified teleconnection (see Supplementary Text).

We calculate the gridded FFWI based on the daily ERA5 reanalysis data and climate modeling outputs from CESM-RESFire and CMIP6 models. The regional average FFWI, BA, NFIRE, and FIREsize for the correlation and composite analysis in Figs. 1, 2, and 4 are estimated using grid-area weighted average over the western U.S. (as shown in Figs. 1a and 2a) based on the reanalysis (FFWI in Figs. 1b and 4a) and modeling data (FFWI, BA, NFIRE and FIREsize in Fig. 2b–e; FFWI in Fig. 4a), respectively. Similarly, the regional average SIC time series in Figs. 1b, 4a, and Supplementary Fig. 2b without detrending and the SIC time series in Supplementary Fig. 1b after detrending are estimated using the same method over the Pacific sector of the Arctic (as shown in Figs. 1a and 2a). Z500i is estimated by

projecting the anomalous geopotential height at 500 hPa in either the reanalysis data or the modeling results onto an identified fire-favorable circulation pattern over the northeastern Pacific and western U.S. (135 °W to 90 °W; 25 °N to 55 °N; as shown in Supplementary Fig. 2a), which is obtained by regressing the ERA5-based anomalous geopotential height at 500 hPa onto the seasonal and regional average FFWI time series over the western U.S. Note this regression-derived fire-favorable pattern in the 500 hPa field resembles the anomalous large-scale circulation pattern favoring regional large wildfires as suggested by a recent study using composite and self-organizing maps analyses[66]. After regional averaging or summation, the time series in Fig. 1b (SIC only), Fig. 4a, Supplementary Fig. 1b, and Supplementary Fig. 2b are also normalized against each other before the composite and correlation analysis. The univariate (1-D) and bivariate (2-D) distribution densities for each index in Fig. 2b-e are then generated and visualized using the Seaborn Python library[67] based on the Kernel Density Estimation (KDE) method, which is a non-parametric way to estimate the probability density function of a random variable.

The bi-monthly Multivariate El Niño-Southern Oscillation index (MEI; positive/negative values correspond to El Niño/La Niña events, respectively)[68] is the time series of the leading combined Empirical Orthogonal Function (EOF) of five different variables (sea level pressure, SST, zonal and meridional components of the surface wind, and outgoing longwave radiation) over the tropical Pacific basin (30 °S to 30 °N; 100 °E to 70 °W). We use MEI for comparing and interpretating the identified S/NP forced responses. The S/NP1 timeseries in both original and detrended data show strong correlations with MEI ($r = 0.96$ with $p$-value < 0.01 in Supplementary Fig. 10; $r = 0.93$ with $p$-value < 0.01 in Supplementary Fig. 12), suggesting this pattern as the most prominent climatic signal is closely related to the coupled ocean-atmosphere conditions in the tropical Pacific.

**Statistical analysis and significance tests.** We first calculate the correlation coefficients between the time series of the regional and seasonal average FFWI (as shown in the red line in Fig. 1b) and the seasonal average Arctic sea-ice concentrations at each grid cell to identify the most sensitive Arctic regions affecting regional fire weather conditions over the western U.S. These SIC-FFWI correlations are shown by the color shading over the Arctic region in Fig. 1a, suggesting the Pacific sector of the Arctic (outlined by the red box over the Arctic in Fig. 1a) as the most critical region of interest. Then the Arctic SIC over this region is averaged to obtain its time series (as shown in the blue line in Fig. 1b) for calculating correlation with the regional average FFWI time series. The correlation coefficient between these two time series is −0.68 with $p$-value < 0.01, reconfirming the close connection between the two regions in terms of temporal variations of the two variables. To exclude the impact of long-term trends in both time series, we also conduct the same analysis based on the detrended time series in Supplementary Fig. 1. The correlation coefficient between the detrended data is lower but still statistically significant ($r = -0.50$; $p$-value < 0.01), suggesting a robust relationship on both interdecadal and interannual time scales.

We then conduct composite analyses based on original or detrended observational and reanalysis data to identify regional weather and fire activity responses to the Arctic sea-ice changes on different time scales. The members in each composite group are selected based on the regional averaged and normalized SIC time series before (Fig. 1b) and after detrending (Supplementary Fig. 1b). We use ±1 standard deviations of the original (detrended) SIC time series as thresholds for selecting members in SIC−/SIC+ (SICnotrd−/SICnotrd+) groups in Fig. 1 (Supplementary Fig. 1). Please note that the MTBS burned area in Supplementary Fig. 1c and meteorological variables such as Z500 and FFWI in Supplementary Fig. 1a, b and air temperature, zonal winds, precipitation, and relative humidity in Fig. 3b, e, h, k are also detrended to keep consistent with the detrended SIC time series in SICnotrd−/SICnotrd+ groups. The long-term trends in time series are removed using the NCL *dtrend* function, while the trends in gridded data are removed by first estimating long-term trends in zonal mean values and then subtracting these trends from gridded values at same latitudes to retain heterogeneity in the zonal direction after removing global warming effects and their footprint in the meridional direction such as AA. In this way, the strong global warming effect is largely removed in the detrended data as shown in Fig. 3b, k.

We use a two-sided Student's $t$-test to test the statistical significance of the differences between two groups of time averaged data (either for each grid cell or for regional averages) such as the seasonal/monthly FFWI composites in Fig. 1a, c ($n = 6$), model-simulated seasonal total burned area composites in Fig. 2a ($n = 40$), model-simulated regional average atmospheric and fire indices in Fig. 2b-e ($n = 40$), and seasonal mean fire weather variables in Fig. 3 ($n = 6$ for reanalysis-based data and $n = 40$ for model-based data). These data generally satisfy the assumptions of the $t$-test including normality and randomness of samples. If a $p$-value obtained from the $t$-test is no larger than a threshold, e.g., $\alpha = 0.01$, 0.05, or 0.1, then we reject the null hypothesis of equal averages in the sample groups at certain significance levels corresponding to the threshold. We also use a stricter control of False Discovery Rate (FDR) method[69] to protecting against overstatement of multiple-testing results due to the influence of possible spatial correlation in Figs. 1a and 2a. Local null hypotheses of each grid cell are rejected if their respective $p$-values are no larger than a threshold level $p^*_{\text{FDR}}$ that depends on

the distribution of the sorted $p$-values:

$$p^*_{\text{FDR}} = \max_{i=1,\dots,N}\Big[ p_{(i)} : p_{(i)} \le (i/N)\,\alpha_{\text{FDR}} \Big] \qquad (5)$$

where $N$ is the number of testing grid cells and $\alpha_{\text{FDR}}$ is the chosen control level for the FDR method. For data grids exhibiting moderate to strong spatial correlation, we can achieve approximately correct global test levels by choosing $\alpha_{\text{FDR}} = 2\alpha_{\text{global}}$ for the FDR method[69]. For the modeling results from different CMIP6 climate modeling systems in Fig. 4 and Supplementary Fig. 9, we compare their consistency in simulated fire weather responses to test the robustness of the climate-fire teleconnection across those models.

To estimate the uncertainty of the probability and intensity changes of extreme burning years, we use a bootstrap method by resampling the model-simulated samples with or without replacement for 10000 times ($n = 10,000$) and then repeatedly estimating the statistics such as the probability and intensity of extreme years based on the resampled data from each experiment (see Supplementary Figs. 5 and 6). By analyzing the newly generated samples from the bootstrap method, we can obtain the statistics of the variables of interest such as the 95% inter-percentile ranges (i.e., percentile values between 2.5% and 97.5%) of the probability of extreme members. These extreme members are defined as modeling years with regional total burned area values above the 95% percentile in the samples from the SICexp+ experiment. Therefore, the probability of extreme years in the SICexp+ experiment ($P^{\text{extreme}}_{\text{SIC exp}+}$) is always 5% in the SICexp+ bootstrap resampling data by definition, while the probability of extreme years in the SICexp− experiments ($P^{\text{extreme}}_{\text{SIC exp}-}$) varies with the mutable resampled data subsets (except the unique subset with the sample size of all 40 modeling years in the non-replacement case). We also test the robustness of these statistical estimates against different ensemble sizes in Supplementary Fig. 6, which shows the probabilities of extreme burning years converging with increasing ensemble sizes of the bootstrap resampling groups. The 40-year ensemble size is large enough to separate the probability of extreme years in the SICexp− experiment ($P^{\text{extreme}}_{\text{SIC exp}-} = 20\%$ with the 95% inter-percentile range of [6.6%, 38.9%]) from that in the SICexp+ experiment even with replacement in the bootstrap resampling processes (i.e., $P^{\text{extreme}}_{\text{SIC exp}+} = 5\%$ is outside of the 95% inter-percentile range of $P^{\text{extreme}}_{\text{SIC exp}-}$). This method has been applied in our previous climate extreme modeling study[63] and other similar applications[70].

The signal-to-noise-maximizing pattern filtering method[37] has been used to separate forced climate responses from each other as well as from climate internal variability. The S/NP filtering method relies on a pattern recognition method named linear discriminant analysis to identify spatial patterns as linear combinations of EOFs that maximize the variance of signal-to-noise ratios among an ensemble of realizations[37]. Here signal is defined by the mean over the ensemble, which consists of the ERA5 reanalysis data and the amip (as listed in Supplementary Table 3) or amip-piForcing (as listed in Supplementary Table 4) models. The ranking of identified S/NP patterns denotes the intensity of the corresponding signal in the ensemble (i.e., signal intensity from high to low with increasing ranking), and the consistency of corresponding temporal variability among different ensemble members denotes the noisiness of the signal. Therefore, the S/NP ranking results are slightly affected by the ensemble composition given different signal-to-noise ratios in each amip or amip-piForcing model, while the spatial distributions of each S/NP are less sensitive to the ensemble composition. Another advantage of this S/NP filtering method over the simple ensemble averaging method is that it greatly reduces the number of ensemble members needed to estimate forced responses by a factor of 7–10 compared to simple ensemble averaging[37]. As suggested by Wills et al.[37], the S/NP-filtered estimate of forced responses based on 3 ensemble members is better than the simple ensemble average of 20 members, and the S/NP-filtered estimate based on 2 ensemble members is only slightly worse. This improved signal-to-noise detection efficiency is of great help for our analyses here given limited numbers of CMIP6 models that provide necessary modeling outputs for the ensemble.

We apply this S/NP filtering method to a multi-variable anomaly field based on seasonal mean surface air temperature, precipitation, FFWI, and Z500 in the Northern Hemisphere from each model. Such combined analysis on all four meteorological fields improves the identification capability and robustness of climate responses to different forcing such as ENSO and AA. After pattern recognition, we calculate the correlation coefficients between timeseries (1981–2014) of these identified S/NPs and climate indices of interest (MEI and SIC in this application) to infer their driving forces. For example, in the S/NP filtering analysis based on the original ERA5 reanalysis and amip modeling data, S/NP1 and S/NP3 show good correlations with MEI ($r = 0.96$; $p$-value < 0.01; Supplementary Fig. 10) and SIC ($r = -0.75$; $p$-value < 0.01; Supplementary Fig. 11), respectively. Therefore, we identify S/NP1 (S/NP3) as the synoptic pattern associated with ENSO (sea ice) variations. Similarly, S/NP1 and S/NP8 show relatively lower but still significant correlations with detrended MEI ($r = 0.93$; $p$-value < 0.01; Supplementary Fig. 12) and detrended SIC ($r = -0.29$; $p$-value = 0.09; Supplementary Fig. 13) in the S/NP filtering analysis based on the detrended ERA5 reanalysis and amip-piForcing data. We then do the composite analysis for each S/NP by comparing differences between the SIC− (SICnotrd−) and SIC+ (SICnotrd+) years to identify forced responses to different climate drivers on different time

scales. The S/NP composite results are shown in Fig. 4d, e in the main text as well as in Supplementary Figs. 10–13 (shown as the ERA5 filtering results derived from the ERA5 and amip model ensemble, or detrended ERA5 and amip-piForcing model ensemble).

We test the robustness of those results by using different CMIP6 experiments (i.e., amip vs. amip-piForcing) or different subsets of the amip models in the ensemble. The MEI-related pattern stays in S/NP1 as the strongest signal across different amip models, while the ranking of the SIC-related pattern varies between the second and the third (e.g., the SIC-related pattern emerges in S/NP2 if using the 5 best-performed amip models (ACCESS-CM2, FGOALS-g3, GFDL-CM4, SAM0-UNICON, and UKESM1-0-LL showing significant negative correlations between SIC and FFWI in Supplementary Fig. 14), while it stays in S/NP3 if using all 15 amip models in the amip model ensemble and drops to the eighth in the detrended amip-piForcing model ensemble). These results suggest a lower signal-to-noise ratio of the Arctic-driven climate effects (as shown in S/NP2 or S/NP3 if using the amip data, or S/NP8 if using the detrended amip-piForcing data) than that of the ENSO-driven ones (as shown in S/NP1) in those CMIP6 models. However, this lower ranking does not necessarily imply that the Arctic-driven effects are weaker than ENSO-driven effects. The relative importance of different climate forcing factors should be inferred by the composite differences for each S/NP, which suggest comparable impacts on regional fire weather by ENSO-driven (Supplementary Fig. 10 and Supplementary Fig. 12) and Arctic-driven (Supplementary Fig. 11 and Supplementary Fig. 13) teleconnection processes in both original and detrended data. For example, the magnitudes of the Arctic-driven changes in S/NP3 (Supplementary Fig. 11) based on the original data with both interannual and interdecadal variability are on the same level of the ENSO-driven changes in S/NP1 (Supplementary Fig. 10). Even when the global warming effects have largely been removed in the detrended ERA5 and amip-piForcing data, those S/NP patterns (S/NPs2-7) with higher rankings than the Arctic-driven one (S/NP8) show subtle differences between the SICnotrd− and SICnotrd+ years, making the Arctic-driven changes in S/NP8 as the second largest contributor to fire-favorable weather conditions over the western U.S. on shorter (interannual) time scales (Supplementary Fig. 13). Moreover, the time series of Arctic-driven patterns in S/NP3 (Supplementary Fig. 11) show a much stronger increasing trend than others on interdecadal time scales, suggesting an increasingly important role of the Arctic changes in modulating regional climate and fire weather over North America. Please refer to Wills et al.[37] for more technical details of the mathematical basis and climate research applications of this method.

**Dynamic and thermodynamic diagnosis.** We examine daily temperature tendencies in CESM-RESFire modeling outputs to quantify the contributions from different dynamic and physical processes to the simulated climate responses. The total temperature tendency (TTEND) consists of two components driven by dynamic processes (DTCORE) and physical processes (PTTEND). The latter can be further decomposed into four major physical processes: moisture processes (DTCOND), longwave heating (QRL), shortwave heating (QRS), and vertical diffusion (DTV). There is one more physical process related to gravity wave drag (TTGW), which is small and negligible in the troposphere of modeling outputs. Therefore, the contribution to the regional average meridional temperature gradient anomaly ($\triangle T = |T_{\text{warm}}| - |T_{\text{cool}}|$; as shown in Supplementary Fig. 7) from each process is given by:

$$\triangle T_{\text{t1}} = \triangle T_{\text{t0}} + \int_{\text{t0}}^{\text{t1}} \text{TTEND}\,dt \qquad (6)$$

where $\text{TTEND} = \text{DTCORE} + \text{PTTEND} = \text{DTCORE} + (\text{DTCOND} + \text{QRL} + \text{QRS} + \text{DTV})$, and $\triangle T_{\text{t0}} = 0$ at the beginning of the sea ice perturbation.

The sign and magnitude of $\triangle T$ depends on the competition of dynamic and physical processes at each timestep. The time evolution of $\triangle T$ shows that it stays above zero in general from September to November (Supplementary Fig. 7), which coincides well with the timing of strong positive fire anomalies (Supplementary Fig. 4).

Besides, note that the meridional temperature gradients remain positive in both the amip and amip-piForcing model ensembles (Supplementary Fig. 9a, e), while the sign and pattern reverse in the difference between these two experiments (Supplementary Fig. 9i). Such results imply that atmospheric and land processes induced by natural and anthropogenic forcing (i.e., GHGs, aerosols, and LULCC) cannot explain the observed circulation and fire weather changes (e.g., Fig. 3a–d) between the SIC− and SIC+ years. These changes are better explained by ENSO- and Arctic-related synoptic patterns according to the S/NP filtering analyses (Supplementary Fig. 15). Please refer to the main text for detailed analysis and interpretation. Also please find all the acronyms used in this study in Supplementary Table 5.

## Data availability

The processed simulation results of the CESM-RESFire sensitivity experiments generated in this study have been deposited at the Figshare website (https://doi.org/10.6084/m9.figshare.13022837.v1). The raw CESM-RESFire modeling input and output data exceed the size limit of the above repository, and are archived on the High

Performance Storage System (HPSS) managed by the National Energy Research Scientific Computing Center (NERSC). They can be obtained by contacting the corresponding authors.

The ERA5 reanalysis data used in this study are distributed by ECMWF from their web site at https://www.ecmwf.int/en/forecasts/datasets/reanalysis-datasets/era5. The ERA5-based FWI is distributed from the Copernicus Climate Data Store at https://cds.climate.copernicus.eu/cdsapp#!/dataset/10.24381/cds.0e89c522?tab=overview. The MERRA-2-based FWI is distributed from the NASA Center for Climate Simulation Data portal at https://portal.nccs.nasa.gov/datashare/GlobalFWI/. The gridded Arctic SIC and SST data are available at https://www.metoffice.gov.uk/hadobs/hadisst/. The bi-monthly MEI.v2 is collected from the Physical Sciences Laboratory of National Oceanic and Atmospheric Administration (NOAA) at https://psl.noaa.gov/enso/mei/. The MTBS burned area data are available at https://www.mtbs.gov/. The CMIP6 model outputs are distributed by the Earth System Grid Federation (ESGF) at https://esgf-node.llnl.gov/search/cmip6/ (see Supplementary Tables 3-4 for model details and references).

## Code availability

The source code for the RESFire model used in this study is available at https://doi.org/10.6084/m9.figshare.7352063. The code for the S/NP filtering analysis is shared and provided by Robert Wills at https://github.com/rcjwills/forced-patterns.

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

## Acknowledgements
This research has been supported by the U.S. Department of Energy (DOE) Office of Science Regional and Global Model Analysis (RGMA) Program area as part of the HiLAT-RASM project. The Pacific Northwest National Laboratory (PNNL) is operated for DOE by Battelle Memorial Institute under contract DE-AC05-76RLO1830. This research used resources of NERSC, a U.S. Department of Energy Office of Science User Facility operated under Contract No. DE-AC02-05CH11231. We thank DOE's RGMA program area, the Data Management program, and NERSC for computational resources to simulate climate sensitivity experiments and to analyze the CMIP6 data. We thank the World Climate Research Programme (which coordinated and promoted CMIP6 through its Working Group on Coupled Modeling), and each contributing climate modeling group for producing and making available their model output. We thank the Earth System Grid Federation (ESGF) for archiving the data and providing access and the multiple funding agencies who support CMIP6 and ESGF. We thank Robert Wills for sharing the code for the S/NP filtering analysis. We thank Jian Lu for helpful discussion to improve the dynamic analysis.

## Author contributions
Y.Z. conceived the research and designed climate sensitivity modeling experiments with P.J.R. and H.W.; Y.Z. conducted the modeling experiments, processed the CMIP6 data, and prepared the first draft of the manuscript. Y.Z., Z.X., and R.Z. analyzed the modeling results, and all authors contributed to the writing of the manuscript.

## Competing interests
The authors declare no competing interests.
