## [Peer Review File · Nature Communications]

REVIEWER COMMENTS

Reviewer #1 (Remarks to the Author):

In this study, the authors highlight the negative correlation between Arctic sea ice in winter and favorable wildfire conditions over the western United States during the subsequent autumn. The study focuses on two main questions: 1) "What is the physical mechanism underlying this Arctic-originated teleconnection suggested by robust statistical correlations on different time scales? 2) How important is this teleconnection comparing to the general global warming effect as well as other teleconnections associated with major climate modes such as ENSO in the past four decades?"

They show that the favorable wildfire conditions are associated with a poleward shift in the North Pacific jet and an increased probability of large wildfires. They also report that this Arctic teleconnection effect on fire weather is comparable in magnitude to ENSO. These conclusions are very interesting, particularly the final one, given the potential to improve seasonal forecasts and help wildfire agencies make better-informed decisions. The manuscript, however, needs some major revisions before it can be accepted for publications.

Major comments

1) The authors have not fully demonstrated that the Arctic teleconnection is commensurate with ENSO. This is associated with the claim "The amplitude of the Arctic-driven fire weather change is of similar magnitude to other leading modes of climate variability such as the El Niño-Southern Oscillation (ENSO) that also influence fire weather in the western U.S."

The conclusion apparently comes from comparing Supplementary Figure 10 (the effect of the S/NP 1 ENSO teleconnection) with Supplementary Figure 11 (the effect of the S/NP 3 Arctic teleconnection), with the authors claiming "comparable impacts on regional fire weather by ENSO-driven ... and Arctic-driven ... teleconnection processes". While there is agreement in the magnitude of the precipitation change over the contiguous US, the change in surface temperature is far greater in the S/NP 3 pattern composite. The aggregate effect of precipitation and temperature changes could be contrasted with composites of FFWI change, as in Fig. 4e for the S/NP 3 pattern, but the corresponding composite for the S/NP 1 pattern is not shown. To assess if the two teleconnections have comparable impacts on fire weather, please include a figure like Fig. 4e but for the S/NP 1 pattern. That said, the authors' results seem to imply that, at least in terms of temperature, the Arctic teleconnection has an even greater impact than ENSO, which seems the more interesting result. Furthermore, the manuscript would benefit from a couple of sentences clearly defining the interpretation of the S/NP rankings e.g. "Given the tight correlation between S/NP 1 and MEI, we identify S/NP 1 as the synoptic pattern associated with ENSO." or something to that effect.

2) The manuscript is very dense with a lot of results. The discussion of methods is unclear in some parts and interpretation of some figures require a lot of effort to understand how the results were computed and what they really mean.

Fig. 1a, for example, contains three results from three different calculations, 2 color bars and no interval contours information. While this might be a compact way of presenting results, one takes a lot of time to figure out what is going on.

3) Additionally, there are other minor concerns regarding the overall structure, interpretation issues with some figures, and language. These issues ought to be easily addressed by the authors and would render the manuscript a valuable addition to the arctic amplification and wildfire literature.

Other comments

1. "rapid declines in Arctic sea ice": Delete the word "rapid". The results pertain to years of sea ice minima and maxima, not the rate of decline.

2. Figs. 1a and 2a in the main text and Figs. 1a and 2a in the Supplementary Information: Essentially the same color bar is used for two different variables in the same figure. Please replace one of the color bars in each figure with a distinct color bar (e.g. purple – green) to clearly

delineate the SIC–FFWI correlation from FFWI difference.

3. Fig 1a: How is the correlation computed in this figure given that the arrays for the red-outlined domains are not co-located spatially and perhaps have different sizes?

4. Fig. 1c: Regarding the last line of the figure caption, what is being tested exactly? Presumably the difference in FFWI between years of sea-ice minima and maxima? Please clarify. The figure caption also states that dot sizes correspond to the 0.05 and 0.1 significance levels of a t-test, but there are three distinct dot sizes. Does the smallest dot size correspond to the difference not being statistically significant? Please clarify. Recommend also that the red and blue legend labels be changed to FFWISIC+ and FFWISIC- (or similar) since they pertain to monthly FFWI.

5. “by prescribing perturbations to the regional Arctic sea-ice concentrations (Fig. 2a)”: For clarification, CESM is run from July to December of a given year with HadISST data perturbed a given percentage as shown in Fig. 2a?

In addition, statistical significance in Fig. 2a is limited to a few points and not over the location of largest differences.

Figs. 2b-e are very complex to interpret with the distributions outside of the panels. How does one interpret the scales for these distributions?

6. “We project the simulated Z500 anomalies onto to a fire-favorable regional circulation pattern to obtain its corresponding time series (hereafter Z500i) in each experiment”: this is not clear what the calculation means.

7. “characterized by increased downward motion, suppressed clouds and precipitation”: Please either delete “increased downward motion” as this is not shown explicitly in Supplementary Figure 3, or reword the sentence to say that suppressed downward motion is highly suggested.

8. Figs. 2b–e. Do (x,y) pairs represent (Z500i, FFWI) seasonally averaged at latitude-longitude grid points from each year in 1981–2010? Are the p-values in the associated text those from comparing the means of the distributions on the y-axes? Please clarify.

9. “These two groups of S/NPs also show constructive contributions by variations in tropical and Arctic surface conditions to regional fire weather changes”: this is not clear. What is meant by “constructive contributions by variations in tropical and Arctic surface conditions”? Please rephrase.

10. Supplementary Figs. 8b and 8c: what are the differences? The figure captions say ERA5.

11. “Supplementary Fig. 10. The first group of forced responses (S/NP1) in seasonal mean multi-field weather variables based on the ERA5 and CMIP6 amip experiment ensemble”: Are the composite differences of S/NP1 in a and b those from ERA5 or CMIP6 amip?

12. BAC equation: the meaning of SICnotrd needs to be explained right after the equation.

13. Given the similarity in results between the ERA5, the detrended ERA5, and CESM sensitivity experiments, what is the utility of the CESM results? What are the model simulations able to highlight that ERA5 cannot? It seems that the CESM results could be forgone without changing the primary conclusions. Doing so would leave more space to motivate and explain the S/NP filtering results, which are complex and answer the second main question in the study.

14. “raising two questions to be addressed in this work: (1) What is the physical mechanism underlying this Arctic-originated teleconnection suggested by robust statistical correlations on different time scales? (2) How important is this teleconnection comparing to the general global warming effect as well as other teleconnections associated with major climate modes such as ENSO in the past four decades?”: Given these primary aims, why is the title about large wildfires?

These questions should be posed to better link with the title or the title should be reworded to reflect the answers to these questions.

15. The manuscript needs structuring for easier readability which would allow readers to better follow the authors’ thought process. As presented, the main text is a very long single section. Recommend breaking the text up into sections according to the methods used and their purpose (e.g. ‘Arctic Correlations’, ‘Impact on Large Wildfire Frequency’, ‘Filtering Teleconnection Patterns’ etc.)

Reviewer #2 (Remarks to the Author):

Increasing Large Wildfires over the Western United States Linked to Diminishing Sea Ice in the Arctic

by Y. Zou, P. Rasch, H. Wang, Z. Xie, and R. Zhang

Overview

This paper describes an extensive investigation of physical mechanisms that explain the reanalysis-based correlation between a fire-weather index over the western U.S. during the main fire season (Sep.-Dec.) and Arctic sea-ice concentrations during summer (Jul.-Oct.) in the Pacific sector of the Arctic Ocean. Associated with this spatial correlation is an anomalous 500 hPa anticyclone over western North America that favors fire-weather conditions, along with an anomalous cyclonic circulation near Alaska. The authors conduct two model simulations using a version of CESM coupled to a regional wildfire model called RESFire in which the sea-ice concentration is prescribed for high and low values in the Pacific sector along with associated SSTs. The difference between these simulations bears a strong resemblance to the correlation patterns exhibited by those based on the reanalysis, suggesting that low sea-ice concentrations in the Beaufort/Chukchi Seas favor fire-weather conditions in western states. This general conclusion is supported by further analysis of various fire metrics (area burned, frequency, fire size) and atmospheric parameters (vertical motions, clouds, precipitation, humidity, winds, and net solar radiation). They find that low sea-ice summers are associated with atmospheric conditions that favor an enhanced fire season via mainly dynamics-related processes. Through additional model simulations for high and low sea-ice concentrations using an AMIP approach -- one with anthropogenic factors included and another with control conditions -- they find further support for the role of sea-ice loss and SST evolution in driving an enhanced fire index. Even further analysis using a signal-to-noise-maximizing pattern (S/NP) filtering method suggests that both sea-ice loss and El Niño (positive MEI) conditions are associated with favorable fire weather, though the ENSO influence has waned while the sea-ice influence has been increasing.

This study makes a substantial contribution to the body of research into the effects of Arctic sea-ice loss on the atmospheric circulation. I have no major criticisms or suggestions for this study, and I

recommend it be published in Nature Communications once the following minor comments are addressed.

Specific comments

1. Another highly relevant study was published in 2017 that should be cited and discussed as to consistencies and differences: Cvijanovic, I. et al, 2017: Future loss of Arctic sea-ice cover could drive a substantial decrease in California's rainfall. Nature Communications, 8, DOI: 10.1038/s41467-017-01907-4.
2. Line 12: make => makes (to agree with nature)
3. 24: What is the definition of a "large wildfire"?
4. 85: reveals => reveal (to agree with changes)
5. Fig. 1a: the red shading indicating positive FFWI correlation over the U.S. appears to have a sharp, artificial discontinuity right along the U.S./Canada border. The authors should look into the causes of this and provide an explanation. I suggest using a blue, not red, box to indicate the study region over the western states in both this figure and also Fig. 2a -- the red one is very difficult to see.
6. 103: Please note inverted scale for SIC.
7. 104: I suggest adding "in a" after "is also shown. I see only one very small dashed contour over western AK. Is this the only area of negative height anomaly?

8. 111-113: This section pertaining to stippling applies only to (a), so I suggest moving it up to the explanation of (a). Other figure captions have similar disjointed descriptions of significance indicators.
9. 140: I suggest starting a new paragraph at "We" to break up this long section.
10. 143-145: I suggest moving the section "...characterized by...western U.S...." up to line 141 after "fire-favorable regional circulation pattern" to help define that circulation pattern.
11. 164: I suggest changing "change" to "difference" to be consistent with main text.
12. 172-173: Are BA and NFIRE measured only for large wildfires?
13. 199: I suggest inserting "Northwest" before coast, as the wetter region is not the whole Pacific coast.
14. Figure 3 j,k,l: It's very difficult to see the RH contours. I suggest using green or some other color.
15. Figure 3c: contour labels missing.
16. 240: I suggest starting a new paragraph at "To..."
17. 255: resembles => resemble
18. 270: I suggest starting a new paragraph at "We..." to break up this long section.
19. Figure 4 caption: I find the nomenclature for the CMIP6/AMIP model runs very confusing. I suggest using something like AMIP-F (for forced) and AMIP-C (for control), and refer to these runs as AMIP rather than CMIP6 model ensemble, as CMIP6 includes many different model configurations.
20. 293: note reversed scale for SIC.
21. 324-327: Could the lower signal-to-noise ratio for the Arctic-driven teleconnection also be due to the fact that sea-ice loss has become substantial (and probably influential) only since the mid-1990s while ENSO has been around forever?
22. 333-334: Similar to previous comment, AA emerged from the noise of natural variability around 2000, and thus has had only a very short period to exert its influence.
23. 349: This statement should be updated to reflect the reality of PAMIP.
24. 354: I suggest inserting at the end of this sentence (after century), "as has the clear decline in Arctic sea-ice coverage."
25. 356: "Autumn" should be replaced with "September".
26. 357: Most models and sea-ice experts predict the Arctic will have a virtually ice-free period earlier than 2050.
27. 359: may sustain => may be sustained
28. 361: Perhaps also mention that homes and infrastructure have expanded into fire-prone areas.
29. 480: Please note that a positive (negative) MEI corresponds to El Niño (La Niña).

Reviewer #3 (Remarks to the Author):

Comments to Authors,

The Manuscript from Zou et.al. provides a long study linking an increase in high fire danger weather occurrence in western United States to diminishing sea ice in the Arctic. There is a tremendous effort at comparing data sets, model experiments, parameterization and weather scenarios to validate the hypothesis and I thank the author for the read and the findings. This use of numerous model runs results in a number of statistically sound tests that support most of the hypothesis (although lacking some clarity to assess with confidence). I found no English mistake or typos, with minor editing required (figures). Overall, these are interesting findings on subject of interest, and I think it does deserve publication after a major revision towards a more concise (and shorter) article that must gain in clarity. My research is mainly in fire behavior modeling and fire meteorology. I have read numerous study on the subject and fire behavior in relation to climate change but this article (method and discussion) is probably one of the most complex given the amount of data sets and scores performed (I probably did not understand fully after several reads), I am sure the same conclusions will have better impact with a clearer and more concise presentation.

Because of the complexity of the study, I do however have some concerns on the overall method for which I may appreciate some answers (first part of review). The second part is a set of possible enhancements to gain in clarity and concision, line specific comments are detailed at the end. Also, there is a mix of metric and imperial system units, but I presume it is because wildfire indices and data are also usually presented this way.

Remarks concerning the method :

1) Fire size. The title states "Increasing Large Wildfires" that may be understood as number of occurrence (l25), likelihood (l352) or size (l269). I guess you probably mean a bit of all these, and it is what is understood from the text, but it does requires clarification from the start. The definition (l379 >1000acres) used as fractional area change (Fig1) between two hypotheses is OK, please make it clearer from the introduction and in the discussion.

2) Fosberg FWI. FFWI, this index is dependent only on instantaneous temperature, relative humidity and wind speed.

This index was given in complement to NFDRS indices, not a part (l442). You use it "in addition to precipitations" l459, you mean you used both or is it some kind composite index ?

You also sometimes refer to FWI (Merra-2), that should not (and you do state it) be compared with, but Supfig 2, is an implicit comparison. FWI is also directly available from the ERA5 reanalysis [FWIERA5], not used, and a more complete (includes drought).

So please better explain also why this (very empirical) index is relevant (not only because it is easy l459), with the little emphasis made on the "fire size" criterion, it may feel that the finding is about the same as the one from Cvijanovic, I. et al.. With a composite index on temperature and moisture. Trough fire size and fire regime, you are drawing a stronger link than just applying fire indices to the studies from [24-28] of your references, but it needs to be clarified (how this study compares to previous Arctic Ice/Dryer Western US studies, maybe just pick one of those).

3) Data sets. MTBS Merra, ERA5, CESM, AMPIs. At some point in the discussion, the article reads as an inter-comparison project where some data sets are invalidated because they are not satisfying an hypothesis. Following the use of all these data sets is complex, What is the very subset of these that support your findings (ERA5, CESM) ?

4) P-Values, It is hard to understand from the first read what data is actually used to compute these values. Monthly means of SIC max/min seasons (I presume n=6 and 40 reanalysis/model refer to selected points in the monthly means). Please clarify so it will be easier to assess, and explain better why these p-values (and the variables selected to compute it) are significant here (not only because you have already used it (l545)).

Possible enhancements, I believe the main article is too long, here are some ideas, but basically

anything that may synthesize the article is welcome.

- Synthetic tables:

There are numerous Acronyms, SIC+nord, Z500, Z500i, CESM-RESFire... some are models, some are experiments, some are data sets. You may compile these in 3 supplementary tables (Experiment, Models and Data sets), describing for each acronym what it is and where it is used.

- General telecommunication diagram:

In your discussion you describe a complex process (high latitude streams, droughts, sea ice loss...) but I believe a large part of the discussion (and maybe some figure) can be synthesized in a figure. I do like as example fig 4 in 27 Cvijanovic, I. et al.

- Focus on the very data sets and model that supports your findings in the main article. The inter comparison is definitely a tremendous work that further corroborates the findings, but they may be referred in a supplementary discussion and supplementary method.

Specific comments :

I56: Why complex interactions cannot be drawn from empirical models ? FFWI is an empirical model of use in your study. I presume you mean that complex interactions are better appreciated with data driven models than with data alone, but clarify statement.

I74: If you are performing a mechanistic evaluation, I would like some kind of diagram somewhere to have isolated components.

I80: We understand later that some observation has been used. Can you introduce where MTBS data is of use (it is understood later that FIRESize is a model output of RESFire).

I92: Why are there no questions about fire weather ? if question 1) is already addressed in [24-28] This is maybe a good place to re-center the problematic on large wildfire.

I97: Fig 1: Better correlations appears to be after year 2000 and in SIC+ years. Maybe a clearer comment.

I204-211: A diagram would be nice

I213 (and I218). This figure is central to the study, especially because we do have a view on precipitations, but it needs to be reformatted to be better understood. Wind vectors are given in m.s.. But no scale or unit found, figures are too small, dots are barely visible in the darker areas and it is difficult to distinguish coastlines from isolines.

I374: Why haven't you used ERA5 provided FWI ?

I396: What are these improvements, and what exactly are the model outputs of use (NFIRE, FIRESize)

I424: Why are these effects neglected, if they exist I presume it makes some sense to include them in a mechanistic approach. If they are impossible to evaluate (and non-representative) given the resolution or area covered you may state it directly in one sentence, or refer directly to your previous study without mentioning that you neglect those effect because it would exert unwanted effects.

I454: Burning index takes essentially fuel into account, it's sub-component, the Spread Index, wind and moisture. FFWI is only air moisture wind and temperature. I do not agree that EMC or FFWI is a simplified or a derivation of BI of NFDRS, you may explain this link better.

I552: If the model fails to reject the hypothesis why are they wrong ? Overall all this inter comparison discussion may be better in a supplementary discussion.

[FWIERA5] Vitolo, C., Di Giuseppe, F., Barnard, C. et al. ERA5-based global meteorological wildfire danger maps. *Sci Data* 7, 216 (2020). <https://doi.org/10.1038/s41597-020-0554-z>

REVIEWER COMMENTS

Reviewer #1 (Remarks to the Author):

In this study, the authors highlight the negative correlation between Arctic sea ice in winter and favorable wildfire conditions over the western United States during the subsequent autumn. The study focuses on two main questions: 1) “What is the physical mechanism underlying this Arctic-originated teleconnection suggested by robust statistical correlations on different time scales? 2) How important is this teleconnection comparing to the general global warming effect as well as other teleconnections associated with major climate modes such as ENSO in the past four decades?”

They show that the favorable wildfire conditions are associated with a poleward shift in the North Pacific jet and an increased probability of large wildfires. They also report that this Arctic teleconnection effect on fire weather is comparable in magnitude to ENSO. These conclusions are very interesting, particularly the final one, given the potential to improve seasonal forecasts and help wildfire agencies make better-informed decisions. The manuscript, however, needs some major revisions before it can be accepted for publications.

Response: Thank you for your comments and helpful suggestions. We have addressed all of your concerns with corresponding changes in the main text and supplementary information to further improve our manuscript. Major changes include revised or new figures (e.g., Fig. 3 and Fig. 5 in the main text, new Supplementary Figs. 15, and more supporting figures in this response document) and revised paragraphs for clarification and comprehensive presentation. We believe these improvements in combination with the responses here have answered your questions about this study. Please see our detailed point-by-point responses (in blue) to the specific comments below. Corresponding changes made in the manuscript are highlighted here in **bold** and marked as tracked changes in the revised manuscript.

Major comments

1) The authors have not fully demonstrated that the Arctic teleconnection is commensurate with ENSO. This is associated with the claim “The amplitude of the Arctic-driven fire weather change is of similar magnitude to other leading modes of climate variability such as the El Niño-Southern Oscillation (ENSO) that also influence fire weather in the western U.S.”

The conclusion apparently comes from comparing Supplementary Figure 10 (the effect of the S/NP 1 ENSO teleconnection) with Supplementary Figure 11 (the effect of the S/NP 3 Arctic teleconnection), with the authors claiming “comparable impacts on regional fire weather by ENSO-driven ... and Arctic-driven ... teleconnection processes”. While there is agreement in the magnitude of the precipitation change over the contiguous US, the change in surface temperature is far greater in the S/NP 3 pattern composite. The aggregate effect of precipitation and temperature changes could be contrasted with composites of FFWI change, as in Fig. 4e for the S/NP 3 pattern, but the corresponding composite for the S/NP 1 pattern is not shown. To assess if the two teleconnections have comparable impacts on fire weather, please include a figure like Fig. 4e but for the S/NP 1 pattern. That said, the authors’ results seem to imply that, at least in terms of temperature, the Arctic teleconnection has an even greater impact than ENSO, which seems the more interesting result. Furthermore, the manuscript would benefit from a couple of

sentences clearly defining the interpretation of the S/NP rankings e.g. “Given the tight correlation between S/NP 1 and MEI, we identify S/NP 1 as the synoptic pattern associated with ENSO.” or something to that effect.

Response: Thank you for the great suggestion. We have added the following sentence in **lines 673-677** of the Method section to clarify how to infer associated climate factors of S/NP patterns:

“For example, in the S/NP filtering analysis based on the original ERA5 reanalysis and amip modeling data, S/NP1 and S/NP3 show good correlations with MEI ($r=0.96$; p value <0.01 ; Supplementary Fig. 10) and SIC ($r=-0.75$; p value <0.01 ; Supplementary Fig. 11), respectively. Therefore, we identify S/NP1 (S/NP3) as the synoptic pattern associated with ENSO (sea ice) variations.”

We have also added a new figure in the supplement (**Supplementary Fig. 15**) to directly compare the composite differences in total and decomposed fields associated with ENSO (S/NP1) and Arctic (S/NP3) changes, which are also shown here as Fig. R1 for your convenience.

Figure R1: The composite differences of fire weather variables between the SIC- and SIC+ years. (a) total difference in precipitation (unit: mm d^{-1}) based on the ERA5 data; (b) total difference in FFWI (unitless) based on the ERA5 data; (c) differences in precipitation associated with the ENSO-related S/NP1 pattern based on the S/NP analysis using ERA5 and 12 amip model ensemble; (d) same as (c) but for differences in FFWI; (e) differences in precipitation associated with the Arctic-related S/NP3 pattern based on the S/NP analysis using ERA5 and 12 amip model ensemble; (f) same as (e) but for differences in FFWI; (g) differences in precipitation as the sum of the ENSO-related S/NP1 pattern and the Arctic-related S/NP3 pattern; (h) same as (g) but for differences in FFWI.

The comparison among total composite difference, ENSO-related difference, and Arctic-related difference suggests that these two climate factors show similar influence on regional fire weather changes in terms of precipitation and FFWI. The Arctic-related composite difference in FFWI (Fig. R1f) is even higher than that related with ENSO (Fig. R1d), which is consistent with the higher impact of Arctic warming on regional surface air temperature as shown in Supplementary

Figs. 10-11 because temperature is a key ingredient for the FFWI calculation (see the Method section in the manuscript).

Furthermore, we calculate the regional average fire weather differences (in terms of precipitation and FFWI) over the western U.S. between the SIC- and SIC+ years based on the S/NP decomposition with different data sources. Table R1 below shows this comparison and reconfirms the critical role of Arctic changes in driving the regional fire weather change, especially when the estimation is based on the ERA5 reanalysis and a 12 member amip model ensemble that captures the correct correlations among Arctic SIC, circulation change (Z500i), and regional fire weather (FFWI) as shown in Supplementary Fig. 14. Adding the other three amip models into this 15-model ensemble for the S/NP filtering analysis reduces the fractional contributions of Arctic-related changes to the total changes due to the introduction of contradictory modeling results from these three models, but it still doesn't change our key conclusion that the Arctic teleconnection plays an important role in modulating regional fire weather and inducing more wildfire hazards over the western U.S.

Table R1 Comparison of regional average fire weather differences based on the S/NP filtering analysis using different data sources

Data source	Regional average differences	PREC (mm d ⁻¹)	FFWI (unitless)
ERA5	Total	-0.15	1.1
ERA5+12-model ensemble	S/NP1 (ENSO-related)	-0.07	0.4
	S/NP3 (Arctic-related)	-0.08	0.5
ERA5+15-model ensemble	S/NP1 (ENSO-related)	-0.08	0.4
	S/NP3 (Arctic-related)	-0.04	0.2

2) The manuscript is very dense with a lot of results. The discussion of methods is unclear in some parts and interpretation of some figures require a lot of effort to understand how the results were computed and what they really mean.

Fig. 1a, for example, contains three results from three different calculations, 2 color bars and no interval contours information. While this might be a compact way of presenting results, one takes a lot of time to figure out what is going on.

Response: We understand the concern here. However, it's challenging to present all the complex processes involved in the Arctic-originated teleconnection in a clear and compact way in a short article format, especially when multiple datasets and analytical methods have been used to cross validate and corroborate each other. Our strategy is similar to other studies, and is achieved by grouping interconnected results based on the same category of data sources (e.g., observations/reanalysis, modeling outputs, etc.) together to support our statements and answers to the specific questions raised in the manuscript. In response to this concern, we have updated the figures and captions and streamlined the narrative to present our methods and results more clearly. Please see these changes in the revised manuscript. Below we also provide additional details associated with the analytical process to help better understanding the whole manuscript.

To summarize briefly, Fig. 1 is designed to show the potential teleconnection between Arctic SIC and regional fire weather (as described by FFWI) and hazards (as described by the MTBS burned area of large wildfires) over the western U.S. based on statistical analyses (e.g., composite analysis; correlation analysis) of observational and reanalysis data, and motivate the

two scientific questions about the underlying physical mechanism as well as its importance in the historical record. We then show modeling results based on CESM-RESFire to directly confirm the teleconnection relation between Arctic sea-ice loss and worsening fire weather that is conducive to more extensive burning in Fig. 2. A more comprehensive comparison between the CESM-RESFire modeling results and ERA5 composite results is then presented in Fig. 3 to illustrate changes in detailed meteorological variables and atmospheric processes affecting regional fire weather. These two figures are used to answer the first question about the physical processes underlying the teleconnection. Figure 4 compares the total fire weather changes and those related with the Arctic forcing according to the S/NP decomposition method based on the reanalysis and CMIP6 modeling data, to address the second question about the importance of this teleconnection in the past four decades. A new schematic diagram has been added as Fig. 5 in the main text to illustrate the key physical processes in the teleconnection. The corresponding diagnostic results for all figures in the main text are shown in the supplementary information.

3) Additionally, there are other minor concerns regarding the overall structure, interpretation issues with some figures, and language. These issues ought to be easily addressed by the authors and would render the manuscript a valuable addition to the arctic amplification and wildfire literature.

Response: Thank you for the helpful suggestions. We've revised the manuscript as noted above, and please see detailed responses below and corresponding changes in the manuscript.

Other comments

1. "rapid declines in Arctic sea ice": Delete the word "rapid". The results pertain to years of sea ice minima and maxima, not the rate of decline.

Response: Done.

2. Figs. 1a and 2a in the main text and Figs. 1a and 2a in the Supplementary Information: Essentially the same color bar is used for two different variables in the same figure. Please replace one of the color bars in each figure with a distinct color bar (e.g. purple – green) to clearly delineate the SIC–FFWI correlation from FFWI difference.

Response: Thank you for the suggestion. We changed the color bars for the SIC-FFWI correlation/SIC difference over the Arctic to purple-green for a better contrast. Please see the updated figures in the revised manuscript and supplementary information.

3. Fig 1a: How is the correlation computed in this figure given that the arrays for the red-outlined domains are not co-located spatially and perhaps have different sizes?

Response: We have revised the manuscript in **lines 575-588** to describe the correlation calculation methods more clearly for the two types of correlations used in Fig. 1.

The first correlation in Fig. 1a is the gridded correlation coefficient between seasonal average (July to October) SIC time series at each grid cell over the Arctic region and regional and seasonal average FFWI time series (the red line in Fig. 1b) to help identifying the most sensitivity Arctic region affecting regional FFWI over the western U.S. in autumn and early winter (September to December). The second correlation in Fig. 1b is the temporal correlation between the time series of regional (the most significant Arctic region in the red outlined box in Fig. 1a as identified by the first step) and seasonal average Arctic SIC (the blue line in Fig. 1b)

and regional and seasonal average FFWI (the red line in Fig. 1b) to demonstrate the close connection between the two regional average variables in terms of their temporal variations.

Lines 575-588 now reads: “We first calculate the correlation coefficients between the time series of the regional and seasonal average FFWI (as shown in the red line in Fig. 1b) and the seasonal average Arctic sea-ice concentrations within each grid cell to identify the most sensitive Arctic regions affecting regional fire weather conditions over the western U.S. These SIC-FFWI correlations are shown by the (green-purple) color shading over the Arctic region in Fig. 1a, suggesting the Pacific sector of the Arctic (outlined by the red box over the Arctic in Fig. 1a) as the most critical region of interest. Then the Arctic SIC over this region is averaged to obtain its time series (as shown in the blue line in Fig. 1b) for calculating correlation with the regional average FFWI time series. The correlation coefficient between these two time series is -0.68 with p-value less than 0.01, reconfirming the close connection between the two regions in terms of temporal variations of the two variables. To exclude the impact of long-term trends in both time series, we also conduct the same analysis based on the detrended time series, shown in the Supplementary Fig. 1. The correlation coefficient between the detrended data is lower but still statistically significant ($r=-0.50$; $p\text{-value}<0.01$), suggesting a robust relationship on both interdecadal and interannual time scales.”

4. Fig. 1c: Regarding the last line of the figure caption, what is being tested exactly? Presumably the difference in FFWI between years of sea-ice minima and maxima? Please clarify. The figure caption also states that dot sizes correspond to the 0.05 and 0.1 significance levels of a t-test, but there are three distinct dot sizes. Does the smallest dot size correspond to the difference not being statistically significant? Please clarify. Recommend also that the red and blue legend labels be changed to FFWISIC+ and FFWISIC- (or similar) since they pertain to monthly FFWI.

Response: We have revised the figure captions in the manuscript and changed the legend labels in Fig.1/Supplementary Figs. 1/2 as suggested to explain that “dot sizes for monthly FFWI in c denote the 0.05 (large), 0.1 (medium), and non-significant (small) significance levels of two-sided t-tests for monthly FFWI differences between years with minimum (FFWI_SIC-) and maximum (FFWI_SIC+) Arctic SIC, respectively.”

5. “by prescribing perturbations to the regional Arctic sea-ice concentrations (Fig. 2a)”: For clarification, CESM is run from July to December of a given year with HadISST data perturbed a given percentage as shown in Fig. 2a?

In addition, statistical significance in Fig. 2a is limited to a few points and not over the location of largest differences.

Figs. 2b-e are very complex to interpret with the distributions outside of the panels. How does one interpret the scales for these distributions?

Response: We have clarified our presentation discussing Fig. 2 in the main text and better explained how to obtain these regional average indices as well as to interpret their KDE-based probability densities in lines 542-563 of the Method section. More details can also be found in the note for the *kdeplot* function (<https://seaborn.pydata.org/generated/seaborn.kdeplot.html>) and the KDE tutorial for joint distribution visualization (<https://seaborn.pydata.org/tutorial/distributions.html>) using the Seaborn library. Briefly,

- We modified **lines 133-140** in the description of the modeling method to clarify that the two CESM-RESFire sensitivity experiments were conducted by replacing the climatological Arctic SIC/SST conditions from July to October in the control run with the multi-year average Arctic SIC/SST conditions of the minimum (SICexp-) and maximum (SICexp+) SIC years, respectively. The modeling results were averaged over the following autumn and early winter (September to December) to examine the teleconnection between Arctic sea-ice loss and regional fire response with a 2-month lag.
- We have also expanded upon our discussion of Fig. 2 (**lines 164-174**) to clarify the significance test results for both gridded values in Fig. 2a and statistical distributions of regional average values in Figs. 2b-e. Please note that the regions showing statistically significant burned area changes do not necessarily correspond to the regions showing largest burned area changes because the ensemble mean values of gridded burned area are heterogeneously distributed over the U.S. (Fig. R2a). Therefore, grid cells with relatively smaller burned area changes could still be statistically significant given lower ensemble mean values of climatological burned area (Fig. R2b).

Figure R2. Spatial distributions of the CESM-RESFire simulated burning activity during autumn and early winter (Sept-Dec) over North America. a. ensemble mean seasonal total burned area (km²) in the SICexp+ experiment; b. seasonal total burned area differences (km²) between the SICexp- and SICexp+ experiments.

- We updated **Figs. 2b-e** by adding Y-axes for the 1-dimensional distributions (i.e., histograms and KDE-based distribution curves) and color bars for the 2-dimensional joint distributions (i.e., KDE-based color shading and contours). The values on the X-axes of those 1-d distributions (corresponding to the variables on the X-/Y-axes of the 2-d joint distributions. For example, in Fig. 2b, the variable on the X-axis of the top horizontal 1-d distribution corresponds to Z500i on the X-axis of the 2-d distribution, while the variable on the X-axis of the right vertical 1-d distribution corresponds to FFWI on the Y-axis of the 2-d distribution) denote standardized variable values in order to quantitatively compare SICexp- and SICexp+ modeling results with statistical significance tests. The Y-values of those 1-d univariate distributions as well as corresponding Z-values of those 2-d bivariate joint distributions denote normalized probability density that are shown qualitatively for visual comparison.

6. “We project the simulated Z500 anomalies onto to a fire-favorable regional circulation pattern to obtain its corresponding time series (hereafter Z500i) in each experiment”: this is not clear what the calculation means.

Response: We have better explained the calculation method in **lines 549-554** of the Methods section in the manuscript. Reader can also refer to the method introduction of teleconnection pattern calculation procedures on the NOAA website (https://www.cpc.ncep.noaa.gov/products/precip/CWlink/daily_ao_index/history/method.shtml) for more technical details.

Briefly, we have adopted the methodology common used for calculating teleconnection indices such as Arctic/Antarctic Oscillation (AO/AAO) by projecting anomalous fields (the Z500 anomalous fields in this case) onto a target spatial pattern/mode (the fire-favorable circulation pattern as shown in Supplementary Fig. 2a). We use this method to calculate the Z500i index as now described in the methods section.

7. “characterized by increased downward motion, suppressed clouds and precipitation”: Please either delete “increased downward motion” as this is not shown explicitly in Supplementary Figure 3, or reword the sentence to say that suppressed downward motion is highly suggested.

Response: This phrase has been deleted as suggested.

8. Figs. 2b–e. Do (x,y) pairs represent (Z500i, FFWI) seasonally averaged at latitude-longitude grid points from each year in 1981–2010? Are the p-values in the associated text those from comparing the means of the distributions on the y-axes? Please clarify.

Response: We have revised the manuscript in **lines 164-174** and **lines 604-613** to clarify the statistical test procedures and their implications.

Essentially, the (x, y) pairs in Figs. 2b-e represent the 2-d joint distributions of the bivariate systems (e.g., (Z500i, FFWI) in Fig. 2b; (FFWI, BA) in Fig. 2c; etc.) simulated in SICexp+ (blue contours) and SICexp- (red shading) with the corresponding univariate density distributions beside the 2-d panels. Each variable has 40 ensemble members from the SICexp+/SICexp- experiments, respectively, that are seasonal (September to December) and regional averaged over the western U.S. (as shown by the cyan outlined box in Fig. 2a). The raw ensemble values of these variables are shown in the scatter plot below (Fig. R3), with their correlation coefficients between each (x, y) pairs listed in Table R2. We chose to show both 1-d distributions of these variables and their 2-d joint distributions because of the causal influence of each x variable on corresponding y variable.

Figure R3. Scatter plots of CESM-RESFire simulated circulation (Z500i), fire weather (FFWI), and burning activity (BA, NFIRE, and FIREsize) variables in the SICexp+/SICexp- experiments. The 2-d joint distributions and 1-d univariate distributions in Figs. 2b-e were generated with the kernel density estimation (https://en.wikipedia.org/wiki/Kernel_density_estimation) method based on these raw samples.

Table R2. Correlation coefficients between each pair of fire-related variables in Figs. 2b-e

r (p-value)	Z500i-FFWI	FFWI-BA	FFWI-NFIRE	FFWI-FIREsize
SICexp+	0.43 (<0.01)	0.45 (<0.01)	0.57 (<0.001)	0.67 (<0.001)
SICexp-	0.47 (<0.01)	0.59 (<0.001)	0.63 (<0.001)	0.59 (<0.001)

Based on the visual inspection of their distributions and the Shapiro-Wilk normality test (<https://docs.scipy.org/doc/scipy/reference/generated/scipy.stats.shapiro.html>), all variables generally meet the assumption of normal distributions with independent samples generated from each modeling year. Meanwhile, each pair of SICexp+/SICexp- results were generated by the same CESM-RESFire model with the same initial conditions but different Arctic SIC/SST boundary conditions (i.e., maximum vs. minimum SIC years). Therefore, we conducted Student's t-tests (https://docs.scipy.org/doc/scipy/reference/generated/scipy.stats.ttest_rel.html) on those "paired" samples for the null hypothesis that those samples from SICexp+/SICexp-

have identical ensemble mean values. If the p-value is smaller than the threshold, e.g., 0.01, 0.05, or 0.1, then we reject the null hypothesis of equal averages. According to the t-test results in the main text, we can confidently reject the null hypothesis at the 0.05 significance level for most variables (except FFWI with p-value of 0.06) and conclude that their ensemble means of the SIC_{exp+}/SIC_{exp-} experiments are not equal.

9. “These two groups of S/NPs also show constructive contributions by variations in tropical and Arctic surface conditions to regional fire weather changes”: this is not clear. What is meant by “constructive contributions by variations in tropical and Arctic surface conditions”? Please rephrase.

Response: We rephrased the sentence in **lines 308-310** of the main text as follows:

“These two groups of S/NPs show consistent responses in regional fire weather with predominant contributions to a warmer and drier western U.S. during the SIC- years by similar magnitudes”.

We also provided additional discussion around Supplementary Figs. 15 in the main text and Fig. R1 in this response.

10. Supplementary Figs. 8b and 8c: what are the differences? The figure captions say ERA5.

Response: Sorry for the confusion. There was a typo here. Supplementary Figs. 8b and 8c are based on the ERA5 and MERRA-2 reanalysis data, respectively. We have now corrected it in the caption.

11. “Supplementary Fig. 10. The first group of forced responses (S/NP1) in seasonal mean multi-field weather variables based on the ERA5 and CMIP6 amip experiment ensemble”: Are the composite differences of S/NP1 in a and b those from ERA5 or CMIP6 amip?

Response: The composite differences of S/NP1 in Supplementary Fig. 10a/b are based on the ERA5 data. Given the high correlation of S/NP1 with the MEI index, the differences associated with S/NP1 can be interpreted as a subset of total differences (as shown in Figs. 3g/j in the main text) related to ENSO variations. The captions of **Supplementary Figs. 10-13** have been updated to clarify this.

We have also added a sentence in **lines 684-685** of the Method section to clarify this:

“shown as the ERA5 filtering results derived from the ERA5 and amip model ensemble, or detrended ERA5 and amip-piForcing model ensemble”

12. BAC equation: the meaning of SIC_{notrd} needs to be explained right after the equation.

Response: The names “SIC_{notrd-}” and “SIC_{notrd+}” are first introduced in **line 149** in the main text to describe the years with minimum (SIC_{notrd-}) and maximum (SIC_{notrd+}) sea-ice concentrations in the detrended SIC time series as shown in Supplementary Fig. 1b.

To clarify this, we have rephrased the sentence in **line 446-448** as follows:

“where \overline{BA}_{SIC-} and \overline{BA}_{SIC+} are the monthly total burned area of large wildfires averaged over the minimum and maximum SIC years in the original (SIC-/SIC+; as shown in Fig. 1b) or detrended (SIC_{notrd-}/SIC_{notrd+}; as shown in Supplementary Fig. 1b) SIC time series, respectively.”

13. Given the similarity in results between the ERA5, the detrended ERA5, and CESM sensitivity experiments, what is the utility of the CESM results? What are the model simulations able to highlight that ERA5 cannot? It seems that the CESM results could be forgone without changing the primary conclusions. Doing so would leave more space to motivate and explain the S/NP filtering results, which are complex and answer the second main question in the study.

Response: We have tried to clarify the scientific questions and our approach in **lines 68-83** and **lines 104-108** of the updated manuscript, and then change our presentation in **lines 134-140** to better explain why we use the CESM-RESFire sensitivity experiments.

The CESM-RESFire climate sensitivity experiments are necessary to answer the first question raised in the manuscript about the physical mechanism of the Arctic-originated teleconnection, while the S/NP filtering method is used to answer the second question about the importance of this teleconnection in the historical observations.

Although the ERA5-based composite differences are similar with the CESM sensitivity results, they can't replace the idealized climate sensitivity experiments because of coexisting variations associated with other climate forcing factors in these reanalysis data. Only the CESM-RESFire modeling experiments provide a direct linkage between the Arctic sea-ice loss and regional fire changes (i.e., fire occurrence/NFIRE, fire size/FIREsize, burned area/BA) given specifically designed modeling settings focusing on the climate impact of sea-ice loss with other fire-related natural and anthropogenic drivers being excluded. The similarity between the modeling results and the ERA5 reanalysis-based composite results (both original and detrended data) in Fig. 3 add more confidence in these modeling results suggesting the significant impact of Arctic sea-ice loss on worsening regional fire weather.

Further evaluation of the importance of the Arctic-originated teleconnection and quantification of contributions from different teleconnections and other sources of climate variability are conducted using the S/NP filtering analysis, which confirms the dominant role of Arctic-related fire weather changes in the observed total changes as shown in Fig. 4.

We think the answers to both questions are equally important in this work and hope the revised manuscript presents these results in a more clear and concise way.

14. "raising two questions to be addressed in this work: (1) What is the physical mechanism underlying this Arctic-originated teleconnection suggested by robust statistical correlations on different time scales? (2) How important is this teleconnection comparing to the general global warming effect as well as other teleconnections associated with major climate modes such as ENSO in the past four decades?": Given these primary aims, why is the title about large wildfires?

These questions should be posed to better link with the title or the title should be reworded to reflect the answers to these questions.

Response: The major objectives of this study are to understand how the teleconnection, as shown by the linkage between the blue shaded blocks in Fig. R4, occur (i.e., the first question in the manuscript) and how important it is comparing to other climate variations and teleconnections (i.e., the second question in the manuscript). Both questions are closely related to large wildfires because the "teleconnection" is established between the Arctic sea-ice loss ("the forcing") and

increasing large wildfires over the western U.S. (“the response”). To clarify this, we have now revised the two questions (as now discussed in **lines 104-108**) as follows:

“(1) What is the physical mechanism underlying this robust teleconnection linking Arctic sea-ice loss with worsening regional fire weather and increasing large wildfires across different time scales? (2) How important is this teleconnection effect on regional fire weather changes during the past four decades comparing to the general global warming effect as well as other teleconnections associated with major climate modes such as the El Niño-Southern Oscillation (ENSO)?”

Fig. R4. A schematic diagram for key factors affecting large wildfires. The blue shaded blocks are involved in the teleconnection as the central focus of this study.

15. The manuscript needs structuring for easier readability which would allow readers to better follow the authors’ thought process. As presented, the main text is a very long single section. Recommend breaking the text up into sections according to the methods used and their purpose (e.g. ‘Arctic Correlations’, ‘Impact on Large Wildfire Frequency’, ‘Filtering Teleconnection Patterns’ etc.)

Response: Thank you for the suggestion. We have restructured the main text by breaking the main text into shorter sections with corresponding section headings to improve the readability of the manuscript. Please see these changes in the revised manuscript for details.

Reviewer #2 (Remarks to the Author):

Overview

This paper describes an extensive investigation of physical mechanisms that explain the reanalysis-based correlation between a fire-weather index over the western U.S. during the main fire season (Sep.-Dec.) and Arctic sea-ice concentrations during summer (Jul.-Oct.) in the Pacific sector of the Arctic Ocean. Associated with this spatial correlation is an anomalous 500 hPa anticyclone over western North America that favors fire-weather conditions, along with an anomalous cyclonic circulation near Alaska. The authors conduct two model simulations using a version of CESM coupled to a regional wildfire model called RESFire in which the sea-ice concentration is prescribed for high and low values in the Pacific sector along with associated SSTs. The difference between these simulations bears a strong resemblance to the correlation patterns exhibited by those based on the reanalysis, suggesting that low sea-ice concentrations in the Beaufort/Chukchi Seas favor fire-weather conditions in western states. This general conclusion is supported by further analysis of various fire metrics (area burned, frequency, fire size) and atmospheric parameters (vertical motions, clouds, precipitation, humidity, winds, and net solar radiation). They find that low sea-ice summers are associated with atmospheric conditions that favor an enhanced fire season via mainly dynamics-related processes. Through additional model simulations for high and low sea-ice concentrations using an AMIP approach -- one with anthropogenic factors included and another with control conditions -- they find further support for the role of sea-ice loss and SST evolution in driving an enhanced fire index. Even further analysis using a signal-to-noise-maximizing pattern (S/NP) filtering method suggests that both sea-ice loss and El Niño (positive MEI) conditions are associated with favorable fire weather, though the ENSO influence has waned while the sea-ice influence has been increasing.

This study makes a substantial contribution to the body of research into the effects of Arctic sea-ice loss on the atmospheric circulation. I have no major criticisms or suggestions for this study, and I recommend it be published in *Nature Communications* once the following minor comments are addressed.

Response: Thank you for your recommendation and constructive suggestions. We have addressed all the specific comments with corresponding changes in the main text to further improve our manuscript. Please see our point-by-point responses (in blue) to your comments below. Corresponding changes made in the manuscript are highlighted here in **bold** and marked as tracked changes in the revised manuscript.

Specific comments

1. Another highly relevant study was published in 2017 that should be cited and discussed as to consistencies and differences: Cvijanovic, I. et al, 2017: Future loss of Arctic sea-ice cover could drive a substantial decrease in California's rainfall. *Nature Communications*, 8, DOI: 10.1038/s41467-017- 01907-4.

Response: We have now cited this pertinent work as the reference #29 with a relevant discussion in **lines 76-83** of the main text.

2. Line 12: make => makes (to agree with nature)

Response: Thank you. This has been corrected.

3. 24: What is the definition of a “large wildfire”?

Response: There are different fire size classes being used in the community (e.g., <https://www.nwcg.gov/term/glossary/size-class-of-fire>). However, there is no uniform criterion for “large wildfire” as discussed here in a general sense.

Since we use the burned area data from the MTBS fire monitoring program (<https://www.usgs.gov/centers/eros/science/monitoring-trends-burn-severity>) in this work, we follow its criterion by focusing on all wildfires that are 1000 acres or greater over the western U.S. from 1984 to 2019. This criterion is now specified in both **line 26** of the introduction and **line 438** with more detailed information of the MTBS dataset in the Methods section.

4. 85: reveals => reveal (to agree with changes)

Response: Thank you. This has been corrected.

5. Fig. 1a: the red shading indicating positive FFWI correlation over the U.S. appears to have a sharp, artificial discontinuity right along the U.S./Canada border. The authors should look into the causes of this and provide an explanation. I suggest using a blue, not red, box to indicate the study region over the western states in both this figure and also Fig. 2a – the red one is very difficult to see.

Response: Thank you for the suggestion. We have changed the outline color for the western US region to cyan in **Fig. 1a, Fig. 2a, Supplementary Fig. 1a, and Supplementary Fig. 2a**.

As shown in the figure below that compares the plots with and without the outlined box over the western U.S., the visual effect of “a sharp and artificial discontinuity” is mainly contributed by the outlined red box used in the old Fig. 1a. After removing this red box or changing its color, the visual effect is largely gone as shown in Fig. R5b. This is also true for other figures (e.g., Fig. 2a, Supplementary Fig. 1a, and Supplementary Fig. 2a) by changing colors of this outlined box. Please also note that these figures were generated based on different data sources (e.g., original/detrended ERA5-based FFWI in Fig. 1a/ Supplementary Fig. 1a; CESM-RESFire simulated burned area in Fig. 2a; MERRA-2-based FWI in Supplementary Fig. 2a) that show different spatial patterns with distinct implications. Please refer to the corresponding discussion in the main text for details of each figure.

Figure R5. Comparison of the visual effect with (a) and without (b) the outlined box over the western U.S.

6. 103: Please note inverted scale for SIC.

Response: Thank you. We have added a note in **line 124** of the updated manuscript:

“note its scale on the left Y-axis is inverted to directly compare temporal variations of both time series”.

The same notes are also added in the captions of **Supplementary Fig. 1** and **Supplementary Fig. 2** in the supplementary information.

7. 104: I suggest adding “in a” after “is also shown. I see only one very small dashed contour over western AK. Is this the only area of negative height anomaly?”

Response: Thank you. The note has been added as suggested.

The negative height anomaly is mainly shown over the western Alaska because this composite difference in Fig. 1a is calculated based on the original ERA5 data without removing the long-term trends. Since the members of the two composite groups (SIC- vs. SIC+) are mainly distributed after (SIC-; see red up-pointing triangles in Fig. 1b) and before 2000 (SIC+; see blue down-pointing triangles in Fig. 1b), their composite differences are largely affected by long-term trends in corresponding variables (Z500 in this case) associated with the global warming effect during this time period.

To remove the effect of long-term trends on composite differences, we also examined and illustrated the composite differences of the same variables after detrending in Supplementary

Fig. 1. By removing the long-term trends in these variables (i.e., Z500, SIC, FFWD, and BA), similar spatial patterns of the composite differences (note that the members of the composite groups are different between Fig. 1 and Supplementary Fig. 1) are largely maintained but more negative height anomalies show up over the boreal regions of the North America continent including both Alaska and Canada. We have further demonstrated that the geopotential height anomaly pattern is robust in the detrended data by doing similar composite analysis but with the S/NP filtered patterns associated with sea-ice changes as shown in Supplementary Fig. 13.

8. 111-113: This section pertaining to stippling applies only to (a), so I suggest moving it up to the explanation of (a). Other figure captions have similar disjointed descriptions of significance indicators.

Response: Thank you. All captions in figures (e.g., **Fig. 1a**, **Fig. 2a**, **Supplementary Fig. 1a**, and **Supplementary Fig. 2a**) with similar descriptions have been revised to address this issue as suggested.

9. 140: I suggest starting a new paragraph at “We” to break up this long section.

Response: Thank you for the suggestion. This paragraph has been broken into two short ones as suggested. Please see this change in **line 162** of the revised manuscript.

10. 143-145: I suggest moving the section “...characterized by...western U.S...” up to line 141 after “fire- favorable regional circulation pattern” to help define that circulation pattern.

Response: Our language here was confusing. The characteristics described after “characterized by...” are associated with the fire-favorable regional circulation pattern. However, these characteristics are not for the fire-favorable pattern itself because this pattern specifically refers to the geopotential height pattern at 500 hPa (Z500) as shown in contours over the northeastern Pacific and western U.S. in Supplementary Fig. 2a. This pattern is obtained by regressing the ERA5-based anomalous geopotential height at 500 hPa onto the seasonal and regional average FFWD time series over the western U.S. (red line in Fig. 1b). Therefore, it represents a circulation configuration that is closely related to regional FFWD variations over the western U.S.

We have now changed “...characterized by...” to “...**concurrent with other regional weather changes including...**” in **lines 167-168** and defined this circulation pattern as well as the calculation of its time series indices in **lines 549-554** of the Method section. We also examined the intensity change of this Z500 pattern (as described by the corresponding Z500i index) between the two CESM-RESFire sensitivity experiments (i.e., SICexp- vs SICexp+) with different Arctic SIC conditions and found that Z500i in SICexp- shows a statistically significant positive shift (p-value=0.01) with respect to SICexp+ (see 1-d KDE-based distributions on the top X-axis of Fig. 2b). This positive shift in Z500i is consistent with a series of fire weather changes that are conducive to large wildfires over this region (see Fig. 2 and Supplementary Fig. 3).

11. 164: I suggest changing “change” to “difference” to be consistent with main text.

Response: Agreed. The “change” here has been changed to **“difference”** to be consistent with the label in the figure.

12. 172-173: Are BA and NFIRE measured only for large wildfires?

Response: To clarify the difference between the model-based and observation-based fire variables, we have now added a note in the caption of **Fig. 2 (lines 204-206)** of the revised manuscript, noting **“that the model-based fire variables are averaged over the coarse model grid cells that describe statistical properties of fire ensembles at each grid cell rather than individual properties of each single fire”**, and pointed out in the Methods section (**lines 488-489** and **lines 545-546**) that “BA” and “NFIRE” in Fig. 2 are based on model simulations instead of observations.

Since the model grid resolution is relatively coarse at a horizontal resolution of 1.9° (latitude) $\times 2.5^\circ$ (longitude) (see lines 466-467 in the Method section), it cannot resolve each single fire within model grid cells. For a large wildfire defined by the MTBS observational dataset (≥ 1000 acres or 4 km^2), it requires a much higher model resolution (at 1-km or 0.01° scale) to explicitly simulate individual large fires. Therefore, the model-based “BA”, “NFIRE”, and “FIREsize” values were measured for “all” fires as direct outputs from the fire model, which represent statistical averages of these variables at $1.9^\circ \times 2.5^\circ$ grid cells.

Although these model-based fire variables cannot be directly compared with their counterparts in the observational data, they are comparable between different modeling experiments as shown in the manuscript. In Fig. 2, all model-based fire variables show positive shifts in SICexp- in comparison with SICexp+, suggesting more and larger fires with increased total burned area under the decreased Arctic sea ice conditions in the SICexp- experiment. This result is generally consistent with the observational MTBS data as shown by the fractional burned area change in Fig. 1c.

13. 199: I suggest inserting “Northwest” before coast, as the wetter region is not the whole Pacific coast.

Response: Thank you. It’s inserted as suggested.

14. Figure 3 j,k,l: It’s very difficult to see the RH contours. I suggest using green or some other color.

Response: The color for the RH contours in **Fig. 3 j-l** has been changed to cyan for a better illustration. Please see the updated figure in the revised manuscript.

15. Figure 3c: contour labels missing.

Response: The contour labels have been added in **Fig. 3c**.

16. 240: I suggest starting a new paragraph at “To...”

Response: Thank you. We break up the long paragraph into three shorter ones to improve its readability. Please see these changes in the revised manuscript.

17. 255: resembles => resemble

Response: corrected.

18. 270: I suggest starting a new paragraph at “We...” to break up this long section.

Response: We now break it up at “**The SST and sea-ice changes...**” as the following analyses and discussion are conducted to answer this question. Please see these changes in **line 300** of the revised manuscript.

19. Figure 4 caption: I find the nomenclature for the CMIP6/AMIP model runs very confusing. I suggest using something like AMIP-F (for forced) and AMIP-C (for control), and refer to these runs as AMIP rather than CMIP6 model ensemble, as CMIP6 includes many different model configurations.

Response: Thank you for the suggestion. We follow the naming convention from the Earth System Documentation (<https://es-doc.org/>) to identify these climate experiments based on the latest generation of Earth system models participating in CMIP Phase 6 (CMIP6). Our terminology has been used to differentiate between the CMIP6 project and its participating models in specific experiments. We now use “amip”/ “amip-piForcing” when referring to the models participating in each CMIP6 experiment, and “CMIP6” when referring to the project itself. We have modified these names throughout the manuscript for clarity in both the main text and figure captions/labels. The models participating in each CMIP6 experiment are now listed in **Supplementary Table 3** (amip) and **Supplementary Table 4** (amip-piForcing).

According to ES-DOC (<https://view.es-doc.org/index.html?renderMethod=id&project=cmip6&id=f1048bbe-5eef-4dda-88ca-d1053cf9ea18&version=1>), “amip” with the long name “Atmospheric Model Intercomparison Project” is a baseline simulation for model evaluation and variability. This “amip” experiment is used as a control for other diagnostic atmospheric experiments such as “amip-piForcing” that is designed to examine whether climate feedbacks during the 20th century are different from those acting on long term climate change and climate sensitivity. Since there are similar experiments such as “AMIP” and “amipPiForcing” from previous Model Intercomparison Project/MIP eras like CMIP5, these new experiments in the CMIP6 era have already been renamed to differentiate from old ones. We tend to use the canonical names for these experiments to avoid adding new difficulties and confusion to our audience.

20. 293: note reversed scale for SIC.

Response: We have added a note in the caption of **Fig. 4** to indicate that the SIC scale on the top left Y-axis is inverted for direct comparison with the other two variables.

21. 324-327: Could the lower signal-to-noise ratio for the Arctic-driven teleconnection also be due to the fact that sea-ice loss has become substantial (and probably influential) only since the mid-1990s while ENSO has been around forever?

Response: We agree that the much stronger long-term decreasing trend in the Arctic SIC, especially after the mid-1990s, might contribute to the growing climate impacts of Arctic sea-ice loss, while the ENSO impacts can be felt throughout the whole time period given relatively stable temporal variations without a significant long-term trend.

However, we also want to emphasize that the “lower signal-to-noise ratio of Arctic-driven teleconnection effects” are derived from the lower ranking number and larger ensemble spread in this Arctic-related pattern across different amip (S/NP3; see Supplementary Fig. 11) and amip-piForcing (S/NP8; see Supplementary Fig. 14) models. These modeling characteristics are distinct from the ENSO-related pattern staying stably in S/NP1 with much smaller ensemble spread either before (see Supplementary Fig. 10) or after detrending (see Supplementary Fig. 12). As “signal” in this method is defined by arithmetic mean over the ensemble including both the reanalysis and the amip or amip-piForcing modeling results, larger discrepancies among individual ensemble members indicate a weaker signal (i.e., showing less agreement with each other). Even though all the amip models are able to reproduce the long-term decreasing trend in the Arctic-related S/NP3 pattern (see time series in Supplementary Fig. 11c), there are still large discrepancies either between the reanalysis and model ensemble mean or among different model ensemble members.

Therefore, we have clarified the distinction between intensity of climate signal (described by the ranking number of each S/NP), noisiness of climate signal (described by the consistency or agreement among each data source), and magnitude of climate effects (described by the composite difference between composite groups) in **lines 650-655 and 686-713**.

22. 333-334: Similar to previous comment, AA emerged from the noise of natural variability around 2000, and thus has had only a very short period to exert its influence.

Response: Agreed. That’s why we designed and conducted the sensitivity experiments using CESM-RESFire to evaluate the climate effects of Arctic sea-ice loss, especially its impacts on regional fire weather and fire activity over the western U.S. These experiments provide more modeling evidence to differentiate Arctic-driven influence from natural variability. We have added a sentence in **lines 134-140** to emphasize this point.

23. 349: This statement should be updated to reflect the reality of PAMIP.

Response: We have updated the status of PAMIP by the following statement in **lines 404-407**:

“more coordinated climate modeling intercomparison projects such as the Polar Amplification Model Intercomparison Project (PAMIP) and the Fire Model Intercomparison Project (FireMIP) have been designed and conducted...”

24. 354: I suggest inserting at the end of this sentence (after century), “as has the clear decline in Arctic sea-ice coverage.”

Response: Revised as suggested.

25. 356: “Autumn” should be replaced with “September”.

Response: Done.

26. 357: Most models and sea-ice experts predict the Arctic will have a virtually ice-free period earlier than 2050.

Response: We have revised the sentence in **line 413** as follows:

“to a sea ice-free Arctic in September before the 2050s”

27. 359: may sustain => may be sustained

Response: Done.

28. 361: Perhaps also mention that homes and infrastructure have expanded into fire-prone areas.

Response: We have revised the sentence in **lines 417-418** by emphasizing the vulnerability of expanding wildland-urban interface (WUI) areas:

“making this region, especially the growing WUI areas, even more susceptible to destructive fire hazards”.

29. 480: Please note that a positive (negative) MEI corresponds to El Niño (La Niña).

Response: We added a note in **lines 564-565**:

“(MEI; positive/negative values correspond to El Niño/La Niña events, respectively)”

Reviewer #3 (Remarks to the Author):

Comments to Authors,

The Manuscript from Zou et.al. provides a long study linking an increase in high fire danger weather occurrence in western United States to diminishing sea ice in the Arctic. There is a tremendous effort at comparing data sets, model experiments, parameterization and weather scenarios to validate the hypothesis and I thank the author for the read and the findings.

This use of numerous model runs results in a number of statistically sound tests that support most of the hypothesis (although lacking some clarity to assess with confidence). I found no English mistake or typos, with minor editing required (figures).

Overall, these are interesting findings on subject of interest, and I think it does deserve publication after a major revision towards a more concise (and shorter) article that must gain in clarity. My research is mainly in fire behavior modeling and fire meteorology. I have read numerous study on the subject and fire behavior in relation to climate change but this article (method and discussion) is probably one of the most complex given the amount of data sets and scores performed (I probably did not understand fully after several reads), I am sure the same conclusions will have better impact with a clearer and more concise presentation.

Because of the complexity of the study, I do however have some concerns on the overall method for which I may appreciate some answers (first part of review). The second part is a set of possible enhancements to gain in clarity and concision, line specific comments are detailed at the end. Also, there is a mix of metric and imperial system units, but I presume it is because wildfire indices and data are also usually presented this way.

Response: Thank you for the constructive comments and helpful suggestions. It's a great challenge to elucidate such a complex teleconnection involving a series of physical processes across multiple components (sea ice, ocean, atmosphere, land) of the Earth system in a relatively short article with a stringent word limit (the main text is limited to 5000 words as required by the journal). We have revised the manuscript extensively to improve the readability and clarity of our work. Please see our point-by-point responses (in blue) to your specific comments below. Corresponding changes made in the manuscript are highlighted here in **bold** and marked as tracked changes in the revised manuscript.

Remarks concerning the method :

1) Fire size. The title states "Increasing Large Wildfires" that may be understood as number of occurrence(125), likelihood(1352) or size (1269). I guess you probably mean a bit of all these, and it is what is understood from the text, but it does requires clarification from the start. The definition (1379 >1000acres) used as fractional area change (Fig1) between two hypotheses is OK, please make it clearer from the introduction and in the discussion.

Response: Yes, the reviewer has correctly understood our intent to communicate that all these measures of fire have increased. We now try to be more explicit about this in the introduction in **line 26** and in the Methods section in **lines 438-442** by noting that both number and burned area of large wildfires in the western U.S. have been increasing over the last four decades as

documented in a new **Supplementary Fig. 16** using the latest MTBS fire data from 1984 to 2019.

2) Fosberg FWI. FFWI, this index is dependent only on instantaneous temperature, relative humidity and wind speed.

This index was given in complement to NFDRS indices, not a part (1442). You use it "in addition to precipitations" 1459, you mean you used both or is it some kind composite index ?

You also sometimes refer to FWI (Merra-2), that should not (and you do state it) be compared with, but Supfig 2, is an implicit comparison. FWI is also directly available from the ERA5 reanalysis [FWIERA5], not used, and a more complete (includes drought).

So please better explain also why this (very empirical) index is relevant (not only because it is easy 1459), with the little emphasis made on the "fire size" criterion, it may feel that the finding is about the same as the one from Cvijanovic, I. et al.. With a composite index on temperature and moisture. Trough fire size and fire regime, you are drawing a stronger link than just applying fire indices to the studies from [24-28] of your references, but it needs to be clarified (how this study compares to previous Arctic Ice/Dryer Western US studies, maybe just pick one of those).

Response: We agree that FFWI is an empirical fire weather index that is simpler than the NFDRS indices or FWI developed in the Canadian Fire Weather Index system, and we have clearly stated that in **lines 518-521** and **lines 533-539** of the method section. However, the relative simplicity of FFWI does not decrease its effectiveness in describing regional fire weather conditions and fire risk/potential in both observational and modeling results, and the use of more sophisticated FWI does not change the key findings as suggested by the analyses done with FFWI in the main text. A major advantage of FFWI over FWI is that the calculation of FFWI does not require hourly meteorological data as FWI does (at 12:00 noon local time; Van Wagner, 1987), and high-frequency data are usually not available from global climate model outputs like CESM and CMIP6 models used in this study. To keep consistency between the reanalysis-based and model-based analyses, we chose to use FFWI for both reanalysis and model results in the main text and examine the robustness of the teleconnection relationship using different fire weather indices and reanalysis datasets in the supplementary text.

We have now revised the introduction and discussion sections of the manuscript (as well as the captions of **Fig. 1** and **Supplementary Fig. 2**) to address this comment, and clarify our use of various fire weather indices derived from different reanalysis datasets (i.e., ERA5; MERRA-2). We have also added additional material and analysis using alternate reanalysis products and fire weather data within the supplementary material (see **lines 17-59** of the **supplementary information**). Briefly, in that section we confirm the robustness of the identified teleconnection and conclude that our results are quite insensitive to the choice of fire weather indices (FFWI and FWI) and reanalysis products (ERA5 and MERRA-2). We now state this in both the main text (**lines 101-103**) and the Method section (**lines 539-540**).

Please note that we have used three different fire weather index datasets (i.e., ERA5-based FFWI in the main text; ERA5-based FWI and MERRA2-based FWI for cross-validation purpose in the supplementary material) to identify the teleconnection between Arctic sea-ice loss and regional fire weather and burning activity change. To reduce redundancy in the manuscript, we show ERA5-based FFWI results in Fig. 1 of the main text, MERRA2-based FWI results in Supplementary Fig. 2, and ERA5-based FWI results in Fig. R7 here to specifically address the

concern raised in this comment. The cross-correlations among all three datasets are also listed in **Supplementary Table 1** for the general audience.

As shown in Fig. R6 below, the teleconnection between Arctic SIC and regional fire weather and burning activity using the new ERA5-based FWI data by Vitolo et al. (2020) is largely consistent with the results shown in our manuscript. The correlation coefficient between SIC and ERA5-based FWI ($r=-0.58$; $p<0.01$) is even higher than that between SIC and MERRA-2-based FWI ($r=-0.37$; $p=0.02$). Therefore, all fire weather datasets confirm our key conclusion about the teleconnection relationship between the decreasing Arctic sea ice and worsening fire weather and increasing large wildfires over the western U.S., which is also corroborated by our CESM-RESFire model sensitivity experiments (Figs. 2/3 in the main text) and the S/NP pattern recognition analysis (Fig. 4 in the main text and Supplementary Figs. 10-14) based on the CMIP6 models participating in the amip and amip-piForcing experiments.

Fig. R6. Observation- and ERA5-based teleconnection between Arctic sea ice and regional fire change. **a**, spatial distributions of the correlation (shading in the Arctic) between seasonal average Arctic sea-ice concentrations in summer and autumn (July to October) and seasonal and regional average FWI over the western U.S. in the following autumn and early winter (September to December), and the difference of seasonal average FWI (shading in the U.S.) between years with minimum (SIC-: red up-pointing triangles in **b**) and maximum (SIC+: blue down-pointing triangles in **b**) Arctic SIC. The difference of seasonal (September to December) average geopotential height at 500 hPa between the SIC- and SIC+ years is also shown in **a** (contours with negative values in dashed lines; unit: m). Stipples in **a** mark regions that are significantly different from 0 at the 0.05 significance level of a two-sided t-test, and hatching in **a** denotes statistically significant regions based on the stricter FDR method (see Methods in the main text) with local gridded p value $\leq p_{FDR}^* = 0.0023$ at the threshold of $\alpha_{FDR}=0.10$. **b**, time series of seasonal and regional average SIC (seasonal mean from July to October; normalized by

its 1981-2010 climatological mean and standard deviation; note its scale on the left Y-axis is inverted to directly compare temporal variations of both time series), FWI (seasonal mean from September to December), and their correlation. The region definitions for the Pacific sector of the Arctic and the western U.S. are outlined by the red and cyan boxes in **a**, respectively. The horizontal dashed lines denote the ± 1 standard deviations of normalized SIC as thresholds for selecting the SIC \pm years. **c**, the composite of monthly FWI (solid lines with dots and error bars) and fractional burned area change of large wildfires (vertical bars) over the western U.S. Error bars in **c** denote ± 1 standard deviations of monthly FWI in each group. Dot sizes for monthly FWI in **c** denote the 0.05 (large), 0.1 (medium), and non-significant (small) significance levels of two-sided t-test for monthly FWI differences between years with minimum (FWI_SIC-) and maximum (FWI_SIC+) Arctic SIC, respectively.

To answer the second question about the comparison of our study with previous ones, we have now added more discussion in the introduction (**lines 73-83**) and discussion (**lines 367-379**) sections of the revised manuscript. Briefly, the previous studies cited in our manuscript and listed by the reviewer mainly focused on the impacts of Arctic sea-ice loss on different regional and seasonal weather changes (e.g., surface air temperature and precipitation) rather than changes in actual burning activity as the central focus of our study. For instance, Cvijanovic et al. (2017) only examined the teleconnection effect of Arctic sea-ice loss on reduced rainfall over California in winter (December to February) that is usually after the fire season in this region. In the past, California's fire season was mainly from May through October, but recent fire disasters have shown that the fire season begins earlier and ends later every year with most susceptible months to large wildfires in autumn (<https://www.frontlinewildfire.com/when-california-fire-season/>). The changing regional fire weather and fire regime agree well with the key findings in our study. There are many other key differences between our study and others, including but not limited to different analytical methods and proposed teleconnection mechanisms as summarized in Table R3 here. These differences are sufficient to distinguish our study from previous ones in that we have not only answered the “*Can it?*” question (the first question in our study) but also the “*Has it?*” question (the second question in our study) as suggested by a previous review paper about ambiguities in research questions regarding the climate impact of Arctic warming (Barnes and Screen, 2015).

Table R3. Key differences between our study and Cvijanovic et al. (2017)

Differences	This study	Cvijanovic et al. (2017)
Research area	The western U.S.	California
Focused season	Autumn and early winter (SOND)	Winter (DJF)
Targeted variable	Fire weather (FFWI, precipitation) and fire variables (fire occurrence, fire size, burned area)	Precipitation
Analytical methods	Idealized climate model sensitivity experiments (to answer the first scientific question or “ Can it? ”) and a S/NP pattern recognition method using reanalysis data and amip/amip-piForcing experiments from CMIP6 (to	Idealized climate model sensitivity experiments only

	answer the second scientific question or “ Has it? ”)	
Proposed mechanism	Massive Arctic sea-ice loss in summer and autumn strongly increases surface absorption of incoming solar radiation through the positive surface albedo feedback, resulting in a strong surface warming and strengthened upward turbulent heat fluxes into the lower troposphere over the sea-ice melted Arctic regions. This surface warming anomaly induces a cyclonic potential vorticity anomaly over the heating source and vicinity land regions like Alaska, which further enhances warm air advection from the ocean into downstream land regions like the western U.S. This dynamic-driven warming anomaly over the western U.S. helps to develop and strengthen an anticyclonic anomaly in the lower to middle troposphere of this region that manifests as a dipole pattern in conjunction with the cyclonic anomaly over the upstream regions like Alaska. This dipole pattern corresponds to a poleward shifted polar jet stream with suppressed precipitation and elevated surface air temperature and vapor pressure deficit over the western U.S., all conducive to increased fuel aridity and more extensive burning during the following autumn and early winter in this region.	Arctic sea-ice loss induced high-latitude energy budget changes propagate into the tropics and trigger tropical circulation and convection response with decreased convection and decreased upper-level divergence in the tropical Pacific. These changes in turn drive a northward-propagating Rossby wave with anticyclonic flow forming in the North Pacific responsible for reduced rainfall in California during winter

3) Data sets. MTBS Merra, ERA5, CESM, AMPIs. At some point in the discussion, the article reads as an inter-comparison project where some data sets are invalidated because they are not satisfying an hypothesis. Following the use of all these data sets is complex, What is the very subset of these that support your findings (ERA5, CESM)?

Response: We used multiple datasets from different sources (observations, reanalysis, climate models) to investigate and validate our proposed hypothesis about the teleconnection between Arctic sea-ice loss and worsened fire weather and increased fire hazards (only available in the observational MTBS fire data and the CESM-RESFire modeling outputs) over the western U.S.

All these datasets were used for the cross-validation purpose, and the results corroborated each other with a consistent conclusion about the validity of this teleconnection mechanism.

A possible confusion may arise from the answer to the second question posed in the manuscript, that is “how important is this teleconnection in the observed history?” This question is different from the first question regarding the physical mechanism of the teleconnection because a theoretically sound mechanism does not necessarily mean it actually occurred with prominent effects in the real world. This is actually one of the key controversies still being debated in the research community about the role of Arctic warming in observed extreme events over mid-latitude regions in recent years (Cohen et al., 2020). To address this problem, we first compared the amip and amip-piForcing experiments to rule out the direct role of GHGs/aerosols/LUC through atmospheric and land processes in driving the observed fire weather changes because these forcing factors resulted in different regional weather changes (see Supplementary Fig. 9i-l) from what we see in the reanalysis data (see the first row of Fig. 3). We then used the S/NP pattern recognition method based on the amip/amip-piForcing data to identify weather patterns associated with specific climate forcing factors such as Arctic SIC (see Supplementary Fig. 11 without detrending or Supplementary Fig. 13 with detrending) and ENSO (see Supplementary Fig. 10 without detrending or Supplementary Fig. 12 with detrending). In this way we can distinguish different variability sources and compare their relative contributions to the observed total fire weather changes as shown in Fig. 4 of the main text and in a new **Supplementary Fig. 15**.

We have now clarified how and why we use all the datasets on **lines 76-83, lines 134-140, lines 271-277, and lines 375-379** of the **main text**, and **lines 17-29** of the **Supplementary Information**.

4) P-Values, It is hard to understand from the first read what data is actually used to compute these values. Monthly means of SIC max/min seasons (I presume $n=6$ and 40 reanalysis/model refer to selected points in the monthly means). Please clarify so it will be easier to assess, and explain better why these p-values (and the variables selected to compute it) are significant here (not only because you have already used it (1545)).

Response: Thank you for the suggestion. We have added short explanations after p -values in the main text (e.g., **lines 164-167**), and updated “**statistical analysis and significance tests**” of the Method section with a more detailed explanation of the statistical analyses and corresponding significance tests.

Briefly, the sampling sizes for each p -value estimation depend on the datasets used for these calculations. For those based on reanalysis data, the sampling size for the correlation coefficient and its p -value estimation is $n=39$ with 37 degrees of freedom following the t -distribution (Fig. 1b), while the sampling size for the composite differences in seasonal (Fig. 1a) and monthly (Fig. 1c) FFWI between SIC- and SIC+ years are $n=6$ as shown by those up- and down-pointing triangles in Fig. 1b. For those based on the CESM-RESFire modeling data, the sampling size for the composite differences in seasonal mean burned area (Fig. 2a/c), FFWI (Fig. 2b-e), fire occurrence (NFIRE in Fig. 2d), fire size (FIREsize in Fig. 2e) between SICexp- and SICexp+ experiments are $n=40$ because we run these model experiments for 40 years. Your understanding in the comment is generally correct.

Possible enhancements, I believe the main article is too long, here are some ideas, but basically anything that may synthesize the article is welcome.

- Synthetic tables:

There are numerous Acronyms, SIC+nord, Z500, Z500i, CESM-RESFire... some are models, some are experiments, some are data sets. You may compile these in 3 supplementary tables (Experiment, Models and Data sets), describing for each acronyms what is it and where it is used.

Response: We agree a table of acronyms will help, and have added a new table in the supplement (**Supplementary Table 5**) for these acronyms used in the manuscript.

- General telecommunication diagram:

In your discussion you describe a complex process (high latitude streams, droughts, sea ice loss...) but I believe a large part of the discussion (and maybe some figure) can be synthesized in a figure. I do like as example fig 4 in 27 Cvijanovic, I. et al.

Response: Thank you for the suggestion.

We have now added a new schematic diagram (**Fig. 5**) in the main text for the description of the whole teleconnection processes initiated by Arctic sea-ice loss in summer and autumn. The mechanism is now briefly introduced in the figure, and discussed in more details on **lines 379-391** of the manuscript. We also add the same figure here as Fig. R7 with corresponding description of the mechanism for your convenience.

“Massive Arctic sea-ice loss in summer and autumn strongly increases surface absorption of incoming solar radiation through the positive surface albedo feedback, resulting in a strong surface warming and strengthened upward turbulent heat fluxes into the lower troposphere over the sea-ice melted Arctic regions. This surface warming anomaly induces a cyclonic potential vorticity anomaly over the heating source and vicinity land regions like Alaska, which further enhances warm air advection from the ocean into downstream land regions like the western U.S. This dynamic-driven warming anomaly over the western U.S. helps to develop and strengthen an anticyclonic anomaly in the lower to middle troposphere of this region that manifests as a dipole pattern in conjunction with the cyclonic anomaly over the upstream regions like Alaska. This dipole pattern corresponds to a poleward shifted polar jet stream with suppressed precipitation and elevated surface air temperature and vapor pressure deficit over the western U.S., all conducive to increased fuel aridity and more extensive burning during the following autumn and early winter in this region.”

Fig. R7. A schematic diagram for the teleconnection between Arctic sea-ice loss and increasing fire hazards over the western U.S. The “L” and “H” denote the cyclonic low pressure and anticyclonic high pressure circulation anomalies, respectively, induced by preceding Arctic sea-ice loss as suggested by the CESM model sensitivity results shown in Fig. 2a. (Background image by NASA/Goddard Space Flight Center Scientific Visualization Studio)

- Focus on the very data sets and model that supports your findings in the main article. The inter comparison is definitely a tremendous work that do further corroborates the findings, but they may be referred in a supplementary discussion and supplementary method.

Response: We now try to deliver our analysis and conclusions more clearly and concisely. We have added more detailed discussion regarding cross-validation of different datasets to the **Supplementary Information**. Please see our detailed response to the previous comments and these changes in the manuscript.

Specific comments :

156: Why complex interactions cannot be drawn from empirical models ? FFWI is an empirical model of use in your study. I presume you mean that complex interactions are better appreciated with data driven models than with data alone, but clarify statement.

Response: We were a little unclear in our messages in the previous version of the manuscript. There is nothing wrong in the use of empirical models (in this case we were specifically referring to the statistical models that use fire weather variables (e.g., air temperature, vapor pressure deficit, precipitation, etc.) to describe potential influence of weather conditions on fire or to predict fire activity like fire occurrence or burned area. We believe our study provides additional

value by: 1) providing a physical explanation for the mechanism that connects sea-ice variability to fire weather; 2) demonstrating that the mechanism operates in CESM-RESFire model experiments designed to exclude all other potential factors; and 3) that the mechanism also appears to operate in many other CMIP6 climate model simulations where many factors are changing fire and weather. Our study uses a number of new analysis techniques and novel simulation strategies to produce these conclusions. We have tried to clarify these points in our revised manuscript on **lines 56-67** in the introduction, and **lines 373-379** in the discussion section.

174: If you are performing a mechanistic evaluation, I would like some kind of diagram somewhere to have isolated components.

Response: We designed and conducted CESM-RESFire sensitivity experiments to obtain a quantitative and mechanistic evaluation of the Arctic impact on regional burning activity. The fire simulation results are shown in Fig. 2 in terms of changes in gridded burned area (Fig. 2a), a fire-favorable circulation pattern (Z500i in Fig. 2b), regional fire weather conditions (FFWI in Fig. 2b-e), and regional fire variables (BA, NFIRE, and FIREsize in Fig. 2c-e) between the SICexp- and SICexp+ experiments. More detailed regional fire weather changes in terms of key fire weather variables (e.g., precipitation, surface air temperature, RH, vapor pressure deficit, etc.) were examined by comparing the CESM-RESFire model results with the reanalysis-based composite results in Fig. 3 and Supplementary Figs. 3/8. A key process inducing more favorable fire weather conditions for large wildfires over the western U.S. was identified in both reanalysis and modeling results that is the poleward shift of polar jet stream as shown in Fig. 3d/e/f. This circulation change is closely related to meridional temperature gradient change following the thermal wind relation, as shown in Fig. 3a/b/c. Therefore, we further decomposed the net temperature gradient change into different temperature tendency terms associated with each contributing processes such as dynamic advection (DTCORE), moisture processes (DTCOND), longwave heating (QRL), shortwave heating (QRS), and vertical diffusion (DTV). This modeling-based diagnostic decomposition result is shown in Supplementary Fig. 7 to provide a mechanistic explanation of how the changed temperature gradient (as shown in Fig. 3c) occurred in the CESM simulation result, which also helps to better understand the observed changes with similar patterns in the reanalysis data (as shown in Fig. 3a/b).

To better demonstrate the sea-ice triggered sequential physical processes affecting regional fire, we have also added a new schematic diagram as **Fig. 5** based on the above modeling results to show critical components and their interrelationship in the teleconnection. Please see this new figure with associated explanation of the proposed physical mechanism **on lines 379-391** of the main text.

180: We understand later that some observation has been used. Can you introduce where MTBS data is of use (it is understood later that FIRESize is a model output of RESFire).

Response: The MTBS fire data are used to calculate the fractional burned area changes in each month between SIC- and SIC+ years as shown by those vertical bars in Fig. 1c.

All the results shown in Fig. 1 are based on observations (effectively MTBS data) or reanalysis data to demonstrate the observed teleconnection relationship between Arctic sea-ice changes and regional fire weather and burning activity. We now try to make this explicit on **lines 85-98**.

192: Why are there no questions about fire weather ? if question 1) is already addressed in [24-28] This is maybe a good place to re-center the problematic on large wildfire.

Response: We have revised our questions to be more explicit about our interest in explaining fire weather (see **lines 104-108**). We believe both questions have not been well addressed in previous studies in the literature, and have revised the introduction (see **lines 73-83**) to highlight that continuous debates about how Arctic warming is influencing midlatitude weather are still ongoing as discussed in a recently published review paper (Cohen et al., 2020).

197: Fig 1: Better correlations appears to be after year 2000 and in SIC+ years. Maybe a clearer comment.

Response: We agree that the relatively short observational records are one of major obstacles to draw any solid conclusion from the pure observation-based analysis, and this motivated the use of climate models to explore possible mechanisms for sea-ice induced fire weather changes. We have revised **lines 367-370** to acknowledge these issues, and motivate our modeling (and others like PAMIP (Smith et al., 2019)) approach to study this problem.

1204-211: A diagram would be nice

Response: We have added a schematic diagram to show these teleconnection processes. Please see this diagram in **Fig. 5** of the main text.

1213 (and 1218). This figure is central to the study, especially because we do have a view on precipitations, but it needs to be reformatted to be better understood. Wind vectors are given in m.s.. But no scale or unit found, figures are too small, dots are barely visible in the darker areas and it is difficult to distinguish coastlines from isolines.

Response: We have updated the figure for a better illustration.

The reference magnitudes for wind vectors are shown on the top-right corner of Fig. 3g/h/i. Dot sizes and coastline thicknesses are increased for highlighting. The color of RH isolines in Fig. 3j/k/l is changed to cyan to better separate from coastlines.

Please see these changes in the revised manuscript.

1374: Why haven't you used ERA5 provided FWI ?

Response: Most observation-/reanalysis-based analyses of this study had been completed in early 2020 before the publication of the ERA5-based FWI dataset by Vitolo et al. (2020) in July 2020.

We appreciate the shared information about this new dataset and we have now repeated the statistical analysis using this data to further confirm the robustness of our results shown in the manuscript. The application of the new ERA5-based FWI data doesn't change our findings (see our previous responses to the second remark for the new result based on ERA5 FWI).

This new ERA-based FWI dataset has been added and discussed in the main text (**lines 429-431**) and Supplementary Information (**lines 30-42**).

1396: What are these improvements, and what exactly are the model outputs of use (NFIRE, FIRESize)

Response: There are many improvements in RESFire compared to the standard CLM4.5 fire model. These improvements are now briefly noted in **lines 458-463** of the main text, with further reference to Table 11 of Zou et al (2019). The model variables used to produce estimates of fire numbers (NFIRE) and fire sizes (FIRESize) are called “NFIRE” and “BA_AVG” in the model output, and there are now notes for these modeling output variables in **lines 486-490** of the manuscript.

1424: Why are these effects neglected, if they exist I presume it makes some sense to include them in a mechanistic approach. If they are impossible to evaluate (and non-representative) given the resolution or area covered you may state it directly in one sentence, or refer directly to your previous study without mentioning that you neglect those effect because it would exert unwanted effects.

Response: Our study attempts to explore the role of sea-ice in regional fire weather changes. That does not mean those fire feedback effects to the climate system are not operating. It just means that they are not required to explain the physical mechanism or the model’s fire responses to sea-ice change of interest. Since we have extensively evaluated those feedback effects in our previous study (Zou et al., 2020), we chose to turn off fire feedback in this study to focus on the specific research objective raised in the manuscript.

We now add a new paragraph with a new **Supplementary Figure 17** to state this explicitly on **lines 492-499** of the method section.

1454: Burning index takes essentially fuel into account, it's sub-component, the Spread Index, wind and moisture. FFWI is only air moisture wind and temperature. I do not agree that EMC or FFWI is a simplified or a derivation of BI of NFDRS, you may explain this link better.

Response: Thank you for the clarification.

As described in the Method section of how FFWI is calculated, this fire weather index is estimated using a series of empirical equations based on surface wind speed, air temperature, and relative humidity with two intermediate variables including the moisture damping coefficient (η) and the equilibrium moisture content (EMC). The calculation of EMC in FFWI is exactly the same as the moisture content calculation in NFDRS that is fundamental to all fuel moisture computations in this system (Cohen and Deeming, 1985).

To clarify this, we have revised the sentence on **lines 529-530**:

“This EMC variable is the same as that used in NFDRS as the fundamental basis to all fuel moisture computations in that more complicated system.”

1552: If the model fails to reject the hypothesis why are they wrong ? Overall all this inter comparison discussion may be better in a supplementary discussion.

Response: We didn't mean to give the impression that these models are wrong, but just that they were unable to capture the correlation features of interest here. We did some simple diagnostic analyses and found much stronger stratospheric responses in the three CMIP6 models listed here, which might be responsible for their failure in capturing the correct climate-fire correlations as observed in the reanalysis data. However, a comprehensive evaluation of distinct modeling performance and inter-model comparisons require much more efforts to draw any solid conclusion on the root cause of these modeling differences, which is apparently out of the scope of this study. Considering the existing level of complexity in the current manuscript, we chose to leave these open questions for a future follow-up study to further explore advantages and disadvantages of different CMIP6 models as well as their modeling performance for this teleconnection.

References

Vitolo, C., Di Giuseppe, F., Barnard, C. et al. ERA5-based global meteorological wildfire danger maps. *Sci Data* 7, 216 (2020).

Van Wagner, C. E. Development and Structure of the Canadian Forest Fire Weather Index System. (Canadian Forest Service, Ottawa, Canada, 1987).

Barnes, E. A. and Screen, J. A.: The impact of Arctic warming on the midlatitude jet-stream: Can it? Has it? Will it?, *WIREs Clim. Change*, 6, 277–286, <https://doi.org/10.1002/wcc.337>, 2015.

Cohen, J. et al. Divergent consensus on Arctic amplification influence on midlatitude severe winter weather. *Nat Clim Change* 10, 20–29, doi:10.1038/s41558-019-0662-y (2020).

Smith, D. M. et al. The Polar Amplification Model Intercomparison Project (PAMIP) contribution to CMIP6: investigating the causes and consequences of polar amplification. *Geosci Model Dev* 12, 1139–1164 (2019).

Cohen, J. D. and Deeming, J. E. The National Fire-Danger Rating System: basic equations. Report No. General Technical Report PSW-82, 16 (P.O. Box 245, Berkeley, California 94701, 1985).

REVIEWERS' COMMENTS

Reviewer #1 (Remarks to the Author):

The authors have done a very good job in improving the original manuscript as well as in clarifying all the previous comments and questions. The paper will be an important contribution to the study of the Arctic amplification and impacts. The manuscript is suitable for publication.

Reviewer #2 (Remarks to the Author):

The authors have done a thorough job of responding to all reviewers' comments and suggestions. I recommend that the manuscript be published.

Reviewer #3 (Remarks to the Author):

Dear Authors, thanks a lot for this revision and detailed answers to questions raised by reviewers.

I did appreciate and understand better this revision, especially the enhanced figures and diagram that added some clarity.

The paper is still hard to read, but understand it is complex as the process it trying to demonstrate so I have now no reserve to provide my acceptance for publication to the editor.